# Estimation of nocturnal $CO_2$ and $N_2O$ soil emissions from changes in surface boundary layer mass storage

Richard H. Grant[1], Rex A. Omonode[1]

[1] Department of Agronomy, Purdue University, West Lafayette, Indiana, 47907, USA

*Correspondence to*: Richard H. Grant (rgrant@purdue.edu)

**Abstract.** Annual budgets of greenhouse and other trace gases requires knowledge of the emissions throughout the year. Unfortunately emissions into the surface boundary layer during stable, calm nocturnal periods are not measureable using most micrometeorological methods due to non-stationarity and uncoupled flow. However, during nocturnal periods with very light winds carbon dioxide ($CO_2$) and nitrous oxide ($N_2O$) frequently accumulates near the surface and this mass accumulation can

be used to determine emissions. Gas concentrations were measured at four heights (one within and three above canopy) and turbulence was measured at three heights above a mature 2.5 m high maize canopy from 23 July to 10 September 2015. Nocturnal $CO_2$ and $N_2O$ fluxes from the canopy were determined using the accumulation of mass within a 6.3 m high control volume and out the top of the control volume within the nocturnal surface boundary layer. Diffusive fluxes were estimated by flux gradient method. The total accumulative and diffusive fluxes during near-calm nights (friction velocities < 0.05 ms$^{-1}$)

averaged 1.16 μmol m$^{-2}$s$^{-1}$ $CO_2$ and 0.53 nmol m$^{-2}$s$^{-1}$ $N_2O$. Fluxes were also measured using chambers. Daily mean $CO_2$ flux determined by the accumulation method were 90% to 130% of those determined using soil chambers. Daily mean $N_2O$ flux determined by the accumulation method were 60% to 80% of that determined using soil chambers. The better signal to noise ratio of the chamber method for $CO_2$ over $N_2O$, non-stationary flow, assumed Schmidt numbers and anemometer tilt were likely contributing reasons for the differences in chambers versus accumulated nocturnal mass flux estimates. Near-surface

$N_2O$ accumulative flux measurements in more homogeneous regions and with greater depth are needed to confirm the conclusion that mass accumulation can be effectively used to estimate soil emissions during nearly calm nights.

## 1 Introduction

Evaluation of the annual emissions of greenhouse and other trace gases emitted from agricultural fields and landscapes requires knowledge of the emissions during representative periods of the year. Micrometeorological methods are widely used to

25 evaluate the emissions and uptake of carbon dioxide ($CO_2$) and to a lesser degree nitrous oxide ($N_2O$). The micrometeorological methods of eddy covariance, eddy diffusion, or Eulerian or Lagrangian dispersion however cannot be used to determine the exchange during stable, calm nocturnal periods due to lack of steady winds and turbulence characteristics assumptions (Pattey, et al, 2002). Due to the non-stationary winds, the integrated horizontal mass flux method is also limited to configurations in which the source area is enclosed or 'fenced' by profile measurements. Various efforts to estimate the exchange during these

periods have been devised- in some cases using purely statistical methods, some using empirical relationships, and some using alternative flux measurement methodologies (Aubinet et al, 2012). The primary difficulties of determining the flux in the surface boundary layer under stable nocturnal conditions include the possibility of advection, non-stationarity of the concentration and velocity fields, and the lack of a similarity theory to describe the non-stationary, intermittent exchange processes. A result of the negligible turbulent transport of mass away from the surface is a temporal change in storage of mass within a layer near the surface primarily a result of low vertical turbulent diffusion. This accumulation occurs initially in a shallow nocturnal surface boundary layer then through light continuous or intermittent turbulence deepens through a thicker (on the order of 100 m) stable nocturnal boundary layer (Kaimal and Finnigan, 1994). Xia et al (2011) noted an accumulation of $^{222}$Rn within a 6.5 m deep surface boundary layer over a grass clearing of a forest preserve during nights with clear sky, light winds, and strong radiative cooling. Similar gas accumulations in the surface boundary layer at night have been conducted for $CO_2$, $CH_4$, and $N_2O$ over pastures and crops (Pattey et al, 2002; Pendall et al., 2010). As weak turbulence mixes the surface boundary layer air with the cooling stable nocturnal boundary layer, gas mass accumulations become evident throughout much of the stable nocturnal boundary layer. Such mass accumulations are reported for $CO_2$, $CH_4$, and $N_2O$ over crops, plantations, and forests (Pattey et al, 2002; Acevedo, et al., 2004; Acevedo, et al., 2008).

Using temporal mass accumulation for estimating flux under stable conditions assumes horizontal mass transport is negligible, there are no local sources of $N_2O$ or $CO_2$ within the control volume, and that the exchange of mass between the control volume and the overlying air is minimal. If there is no flow in the surface boundary layer (SBL), then gases emitted from the soil surface will diffuse upward at roughly the rate of molecular diffusion (approx. $10^{-5}$ m$^2$s$^{-1}$). Such conditions are approximated in soil flux chambers but do not occur in the surface boundary layer beyond the laminar layer at the surface. Compared to the typical turbulent diffusion exchange coefficients, the molecular diffusion rate is negligible. Consequently gas diffusion from the surface is effectively stopped at any altitude were the diffusion rate decreases a few orders of magnitude. This provides the effective 'cap' on the mixing of gases in the control volume layer.

Many approaches have been used to define the conditions in which the accumulation of a gas as effectively capped in the surface boundary layer. Since the friction velocity (u$_*$) provides an index of turbulent mixing, Pattey et al (2002) used a u$_*$ threshold for validating the quality of the 'cap'. Pendall et al (2010) defined the top of the accumulation control volume based on significant correlations between $CO_2$ (presumed from soil respiration) and CO, $CH_4$, $N_2O$, and $H_2$. The top of the control volume has been estimated by Acevedo et al (2004) using the top of an observed fog layer or the height of constant potential temperature and specific moisture in the early morning. Acevedo et al (2008) used the height of the strongest potential temperature inversion as the control volume top. Pattey et al (2002) determined the accumulation over the entire 10 m of profile measurements under constrained turbulent flow conditions. Using these 'cap' definitions, the temporal change in mass accumulations have been determined over relatively thin layers of air over crops (10 m thick; Pattey et al, 2002), pastures (5 m thick; Pendall et al., 2010) and plantations (8 m thick; Pendall et al., 2010). Other much thicker layers of at least 20 m have been defined over forests (Acevedo, et al., 2004; Acevedo, et al., 2008; Pendall et al., 2010).

The source area represented by a flux measurement (termed 'footprint') has a turbulent and advective component (Vesala et al, 2007) and depends on the height of the measurements, the duration of the averaging period used in the method, and the flow conditions during the measurements. The turbulent component to the flux footprint also cannot be readily assessed under these complex flow conditions since the determination of turbulent flux footprints depends on definable (stationary) flow (Vesala et al, 2007). Furthermore, the typical mass accumulation micrometeorological method integration period of at least an hour is typically longer than the averaging period of eddy covariance and most other micrometeorological flux measurement methods and hence less likely to be stationary flow throughout the period. If the flow were stationary, the longer integration period for the accumulation flux over other micrometeorological flux methods such as eddy covariance results in larger represented source areas for the measured flux than other micrometeorological flux measurements. Given the complex flow conditions of the stable nocturnal SBL (non-stationary flow and low turbulence), these longer integration intervals will result in an increased potential for advective mass contributions contributed to the SBL by nearby sources with differing emissions. Chambers et al (2011) attempted to determine the relative contribution of $^{222}$Rn accumulation in the atmospheric boundary layer to a height of 50 m from mixing of local sources and that advected from 'remote' regions with greater or less soil flux using seasonal average HYSPLIT simulations (Draxler and Hess, 1998). Chambers et al (2011) found a frequent decoupling of flow between the 2 m and 50 m measurement heights during the night and suggested that the 50 m flow was in the residual layer and not the surface boundary layer. Such decoupling would not be simulated by HYSPLIT resulting in potentially significant errors in nocturnal surface boundary layer footprint determination. Furthermore, since HYSPLIT relies on well-defined turbulent characteristics to model the back-trajectory, the poorly defined non-stationary, intermittent flow of the nocturnal boundary layer cannot be well-represented in any HYSPLIT-based fetch estimate. Consequently estimates of the flux footprint during stable nocturnal conditions were not estimated by HYSPLIT but by assuming stationarity of hourly mean flow for 2 to 12 hours with footprint estimates driven by atmospheric motion in the daily, not turbulence time scales. These assumptions resulted in footprints extending 10 to 40 km away for winds averaging approximately 1.5 ms$^{-1}$ (Chambers et al, 2011). Biraud et al (2002) estimated the fooprint for their $^{222}$Rn flux estimates for the atmospheric boundary layer (ABL) based on sampling at 20 m and an assumed well-mixed ABL using HYSPLIT and assumed steady winds over the entire ABL for multiple days. They assumed the trajectory air parcel was in contact with the land if it was within 2 km above ground level (within the ABL). Consequently their footprint estimates were driven by atmospheric motion above the stable nocturnal surface boundary layer. As Chambers et al.(2011), Biraud et al (2002) assumed stationary flow conditions at all times and found wind speeds (at 10 m) ranging from 1.5 ms$^{-1}$ to 8 ms$^{-1}$ over four days corresponded to footprints extending 150 km to 200 km from the measurement location.

We evaluated the nocturnal flux of $CO_2$ and $N_2O$ from maize-cropped land based on the temporal accumulation of mass storage within the surface boundary layer constrained vertically by the flow characteristics at the top of a layer 6.3 m deep.

## 2 Methods

$N_2O$ and $CO_2$ fluxes were measured using three methods during the night between 2000 and 0400 local time (hereafter referred to as 20 h LT and 04 h LT) over nitrogen-fertilized fields during the summer of 2015. These fields are located in a relatively flat and homogeneous terrain (Fig. 1a) near West Lafayette, Indiana, USA (40.495° latitude and -86.994 ° longitude). The terrain rises to the north at a rate of only 2 m $km^{-1}$ and land use is predominantly agricultural with cropped land covering 100% of the land within 1 $km^2$ and 97% of the within 10 $km^2$ (Table 1) and 83% within 25 $km^2$. Crops are generally alternating between maize and soybean with 83%, (1 $km^2$) 46% (10 $km^2$) and 40% (25 $km^2$) in maize in 2015.

The instrumented towers (described below) were situated in a tilled field ('200Sp'; Fig. 1b) in which 220 kg N $ha^{-1}$ were applied as anhydrous ammonia (AA) at pre-plant in spring 2015. Three other fertilizer treatments were applied in fields near the towers: a 220 kg N $ha^{-1}$ AA application on a no-till field ('200Fa; Fig. 1b), and a 110 kg N $ha^{-1}$ AA application during the fall of 2014 followed by a pre-plant spring AA application of 110 kg N $ha^{-1}$ ('100Fa/100Sp'; Fig. 1b) on a tilled (north) and no-till (south) field.

$N_2O$ and $CO_2$ concentrations were measured from air sampled out of a 7 L $min^{-1}$ air flow drawn from 1$\mu$m-filtered inlets at three heights: 2.8 m, 5 m, and 8 m above ground level (agl). Air was sampled sequentially for 5 minutes at each inlet. Mean concentrations were based on the last three of each five-minute interval to account for the time lag associated with the air flow and the measuring instruments. The 2.8 m point sample was made from a mast that was 18 m from the 5 and 8 m measurement mast (Fig. 1b). In addition a line sample based on a 50-m line with ten inlets drew air at 1 m within the canopy (Grant and Boehm, 2015). The 1 m in-canopy line sample measurement was positioned between 50 m and 25 m (line sample end to end) from the 5 m and 8 m single point mast measurements (Fig. 1b). The 2.8 m single point measurement was made between 45 m and 65 m from the 1-m line sample (end to end) and 18 m from the 5 and 8 m measurement mast (Fig. 1b). The $N_2O$ in the sampled air was measured using an IRIS 4600 difference frequency generation (DFG) laser mid-infrared (IR) analyzer (ThermoFischer Scientific, Franklin, MA) with a measured $N_2O$ minimum detection limit (MDL; 3 sigma) of 0.3 nmol $mol^{-1}$. The $CO_2$ in the sampled air was measured using a LiCOR 840 non-dispersive IR analyzer (LiCOR, Inc., Lincoln, NE) with a measured $CO_2$ MDL of 5 $\mu$mol $mol^{-1}$. The moisture content of the sampled air was also determined by the LiCOR 840 non-dispersive IR analyzer. All concentrations were corrected to dry air.

Atmospheric pressure, temperature and relative humidity were measured at 2.5 m at 5-min intervals on a weather station within 100 m of the gas measurements. Turbulence was measured at three heights (2.5 m, 5 m, and 8 m) using a 3-dimensional sonic anemometer (RM Young 81000, RM Young, Inc., Traverse City, MI). Turbulence was sampled at 16Hz and recorded at 10Hz. The minimum detection limit (MDL) was approximately 0.01 $ms^{-1}$. Since the tethered towers was tilted but shifted slightly in tilt due to shifts in the wind direction, a double rotation rather than planar rotation was made to correct the flow coordinate system for each 30-min turbulence-averaging interval (Lee et al, 2004). The MDL for the friction velocity ($u_*$), based on error propagation of the anemometer MDL through the definition of $u_*$, was estimated to be 0.014 $ms^{-1}$. In reality, the non-stationary

flow conditions in the stable nocturnal boundary layer results in sensitivity of $u_*$ on the specific averaging period and consequently is more uncertain than the calculated MDL. Stability was assessed using the local Obukhov length ($\Lambda$) based on local measures of heat and momentum transfer within the stable boundary layer (van de Wiel et al, 2008).

The accumulation of constituent C ($Q_{accum,c}$) over the maize canopy was based on gas concentration measurements (using the DFG and NDIR instruments) made at three heights (2.8m, 5m and 8m; Fig. 1b) on an 8m tower and one height representing an integrated line concentration in the maize canopy (1 m; Fig. 1b). Accumulation flux was determined into the layer according to:

$$Q_{accum,C} = \frac{\Delta \int_0^{6.3} C dz}{\Delta t}$$

(1)

using Newtonian integration and assuming the concentration of C ($CO_2$ and $N_2O$) between the ground and 1 m was constant and equal to that at 1 m . The mass accumulative flux was calculated as the linear slope of the time resolved accumulation of three measurements over 90 minutes. Turbulent conditions were segregated into those with $u_*$ less than or greater than or equal to 0.05 $ms^{-1}$ (approximately four times the estimated sensor MDL of 0.014 $ms^{-1}$). This threshold was lower than that used by Pattey et al (2002), who used a threshold of 0.1 $ms^{-1}$ for both $u_*$ and standard deviation of w ($\sigma_w$) to estimate flux by mass accumulation and lower than that used by Wagner-Riddle et al (2007) who used a $u_*$ threshold of 0.1 $ms^{-1}$ and a Monin Obukhov stability of less than 2 to estimate flux by flux gradient during stable boundary conditions.

The diffusive flux of constituent C ($Q_{diff,c}$) out the top of the control volume (6.3 $m^3$) under both unstable and stable conditions was determined using the flux gradient method:

$$Q_{diff,C} = K_C \frac{\Delta C}{\Delta z}$$

(2)

where the concentration gradient ($\Delta C/\Delta z$) was calculated above the canopy between 5 m and 8 m (van de Wiel et al, 2008). The $\Delta C$ MDL was estimated at 12.7 $\mu mol\ mol^{-1}$ for $CO_2$ and 0.5 $nmol\ mol^{-1}$ for $N_2O$ based on the MDL for the respective gas concentrations. The eddy exchange coefficient ($K_c$) for the top of the control volume was determined using 3D sonic anemometer measurements at 5m and 8m using the similarity method of Schaefer et al. (2012) and the molecular Schmidt number (0.91 for $CO_2$ and 0.95 for $N_2O$; Massman, 1998). The molecular Schmidt number was used in place of the preferred turbulent Schmidt number because no independent measure of the coefficient was possible in this experiment and literature values for the turbulent Schmidt number are quite variable (Flesch et al, 2002). Given the sonic anemometer measurement error in wind speed and the corresponding error (based on theoretical error propagation) in $u_*$, the error in $K_c$ was estimated at 22%, or approximately 0.0035 $m^2s^{-1}$. Given the MDL of a $\Delta C$ of 18 $\mu mol\ mol^{-1}$ $CO_2$, the MDL of $\Delta CO_2/\Delta z$ at 6.3 m (top of control volume) was estimated at 0.2 $\mu mol\ m^{-4}$. Given the MDL of a $\Delta C$ of 0.6 $\mu mol\ mol^{-1}$ $N_2O$, the MDL of $\Delta N_2O/\Delta z$ at 6.3 m was estimated at 7.0 $nmol\ m^{-4}$. Diffusive fluxes where the $\Delta C$ or $K_c$ were less than the MDL were invalidated. The MDL

of the diffusive flux (Eq. 2), based on theoretical error propagation, of $CO_2$ and $N_2O$ were estimated at 0.7 nmol m$^{-2}$s$^{-1}$ and 0.02 nmol m$^{-2}$s$^{-1}$ respectively. As previously stated, the non-stationary flow conditions in the nocturnal surface boundary layer result in greater uncertainty in u$_*$ and $K_c$ than calculated by theoretical error propagation. In addition, since the double rotation coordinate tilt induce additional errors in u$_*$ for u$_*$ less than 0.15 ms$^{-1}$ (Foken et al, 2004), the error in $K_c$ was expected to be much lager for low turbulence conditions. Diffusive fluxes were determined over 30-min averaging time intervals. All sampling periods with invalid (below theoretical MDL) diffusive fluxes were set to zero. Z-less flow (Mahrt, 2011) was assumed to not be present at the stable control volume top: if present the method of diffusive flux calculation (Eq. 2) would be invalid.

Total nocturnal fluxes of $CO_2$ and $N_2O$ over 90-minute intervals were determined by adding the 1.5 h mean of the diffusive flux (Eq. 2) to the accumulative flux (Eq. 1). Calculated fluxes were further screened for extreme outliers: values greater than ten times the standard deviation of the flux were excluded from analysis.

The $CO_2$ and $N_2O$ emissions were also determined using the vented static chamber method (Mosier et al, 2006). Measurements were made between 10 h LT and 14 h LT in the 200Fa, 100Sp/100Fa and no-N treatment fields (Fig. 1). Measurements were made generally at 23 h LT in the 200Sp treatment area where the masts were located except for a four day period (5-8 August 2015) in which the diurnal variation in chamber $N_2O$ emissions were assessed when measurements made at 00, 06, 12, and 18 h LT and measurements made at 13 h LT on 24 and 28 July. The chamber consisted of aluminium anchors (~0.74 by 0.35 by 0.12 m) driven about 0.10 m into the soil; at each sampling time lids covered the anchors to result in a chamber volume of approximately 32.4 L. On each sampling date, gas samples were collected from the chamber headspace through a rubber septum at 0, 10, 20, and 30 min after chamber deployment using a gastight syringe, and then transferred into pre-evacuated 12 mL Exetainer vials (Labco, High Wycombe, UK). Nitrous oxide and $CO_2$ concentrations of the gas samples were determined using a gas chromatograph (Varian 3800 GC, Mississauga, Canada) equipped with an automatic Combi-Pal injection system (Varian, Mississauga, Canada). Fluxes were calculated from the rate of change of the $N_2O$ concentration in the chamber headspace assuming a linear rate of change in concentration within the headspace. The MDL determined based on the 99% confidence interval of the rate of change was 3.7 nmol m$^{-2}$s$^{-1}$ for $CO_2$ flux and 0.7 nmol m$^{-2}$s$^{-1}$ for $N_2O$.

Comparisons between the daily mean chamber flux and mass accumulation flux were made over three time intervals: 22 to 31 July, 1 to 22 August, and 23 August to 2 September. All valid flux measurements (chamber or accumulation) for a given day were averaged to estimate the day's flux. Only chamber measurements made in the field where the accumulation measurement were made are included. Statistics of mass accumulation measurements were made regardless of the time of day of measurement. Student's t-test was used to determine if there was a significant difference at p=0.05 between the chamber and mass accumulation measurements.

The potential influence of advection of $CO_2$ and $N_2O$ from the surrounding landscape on the accumulated masses at the research site was evaluated based on 2015 land use and typical fluxes given the land use. Land use during the 2015 growing season

was assessed using CropScape Cropland Data Layer (USDA, 2017). Dominant land use, excluding developed land, was assessed for the surrounding 1 km$^2$ and 10 km$^2$ area of the measurement tower (Table 1). Fluxes associated with each land use were selected from the literature based on similarity of soil type (research site: Drummer silty clay loam), land management (research site and surrounding field tile drained, chisel plow vs. no-till with various fertilization rates and fertilizer type) and crop phenological stage (research site: maturity for soybean and maize) (Table 1). In addition, literature-reported fluxes derived using micrometeorological approaches were preferred over fluxes derived from soil chambers unless specifically reporting soil+root fluxes.

## 3 Results and Discussion

Measurements were made over the period 23 July to 11 September, 2015 resulting in 1685 30-min averaged records. Within this period there were 600 ½ h periods with $N_2O$ measurements and 370 30-min periods with $CO_2$ measurements between 19 and 03 h LT. During this period, the mature maize canopy was 2.5 m tall (H).

### 3.1 Near-surface layer profiles

A common feature of the nocturnal $CO_2$ and $N_2O$ concentration profiles is an increase in concentration near the surface over time (Fig. 2b,c). Mass accumulations of $CO_2$ and $N_2O$ were observed over the mature maize canopy when wind speeds were low at 8 m (3.2H) (Fig. 2a). The increased concentrations were assumed to be a result of gaseous emissions largely from the soil surface. Mean wind speed (U) and the ratio of variability in w ($\sigma_w$) to $u_*$ at both 5 m and 8 m were significantly lower when $u_* < 0.05$ ms$^{-1}$ than when $u_* > 0.05$ ms$^{-1}$ (Table 2; Fig. 3). Over the nocturnal period of 19 to 07 h LT, the averaged local stability at 8 m (z/$\Lambda$; van de Wiel et al, 2008) was positive regardless of $u_*$ between 19 and 03 h LT and negative from 0300 and 0700 h LT. The negative stability expressed the influence of dawn occurring around 05 h LT (Table 2). Stable conditions (positive $\Lambda$) at 8 m occurred during 28% of the measurement periods (465 30-min measurement intervals).

Sonic temperature ($T_s$) increased with height between 3 and 5 m under low turbulent conditions throughout the night while increasing turbulence between 20 and 07 h LT shifted the $T_s$ gradient from positive to negative with height (Fig. 2). However at the top of the measured profile, the temperature gradient was nearly zero for $u_* < 0.05$ ms$^{-1}$ (Table 3). The mean bulk Richardson number ($R_B$) at the geometric mean height of the top two measurements averaged 2.3 when $u_* < 0.05$ ms$^{-1}$. For conditions with $u_* >= 0.05$ ms$^{-1}$ the mean $R_B$ was -1.2. Shifts in wind direction above the canopy (5 to 8 m height) were highly variable for $u_*$ less than approximately 0.05 ms$^{-1}$ (Fig. 3). These shifts coincided with vertical wind velocity variance less than 0.01 m$^2$s$^{-2}$ and the horizontal wind velocity variance less than 0.1 m$^2$s$^{-2}$ (Fig. 3). At these low turbulence conditions, turbulent transport of gases originating at the earth surface is minimal resulting in the accumulation of gases in a layer of air bounded

by a 'cap' in the surface boundary layer. The top of the surface-influenced control volume in which mass accumulation was set at 6.3 m (geometric mean of 5 m and 8 m; 2.5H) (Fig. 4).

Over the 19 to 07 h LT timeframe, the line-averaged concentrations of $CO_2$ at 1 m within the canopy ranged from 354 μmol mol$^{-1}$ to 1038 μmol mol$^{-1}$ while point concentrations at 8 m agl (5.2 m or 2.9 H above the canopy) varied from 358 μmol mol$^{-1}$ to 862 μmol mol$^{-1}$. The difference between the 5 m (1.7 H) and 8 m (2.9H) $CO_2$ concentrations ranged from -11.4 μmol mol$^{-1}$ to 337 μmol mol$^{-1}$. Eleven percent of the 90 minute mean concentration gradients at the top of the layer were high enough to calculate a turbulent diffusive flux. The mean $CO_2$ gradient ($\Delta CO_2/\Delta z$) was less than or equal to the MDL when $u_* > 0.05$ ms$^{-1}$ (Table 3).

Over the 19 to 07 h LT timeframe, the line-averaged $N_2O$ concentrations within the canopy (0.4H) ranged from 0.313 μmol mol$^{-1}$ to 0.467 μmol mol$^{-1}$ while the point sample at 8 m ranged from 0.295 μmol mol$^{-1}$ to 0.448 μmol mol$^{-1}$. The difference between the 5 m (1.7 H) and 8 m (2.9H) $N_2O$ concentrations above the canopy ranged from -0.357 μmol mol$^{-1}$ to 0.059 μmol mol$^{-1}$. Twelve percent of the 90 minute mean concentration gradients at the top of the layer were high enough to calculate a turbulent diffusive flux. The mean $N_2O$ gradient ($\Delta N_2O/\Delta z$) was less than the MDL when $u_* > 0.05$ ms$^{-1}$ (Table 3).

A common feature of the mean concentration profiles of both $CO_2$ and $N_2O$ was a lower mean concentration from air sampled at a point 3 m (1.2H) than both the 1 m (0.4H) and 5 m (1.7H) mean concentrations. This may be a result of the close proximity of the 1.2 H point measurement to the canopy top representing only local canopy conditions. Conversely, the spatially-averaged line concentration in the canopy at 0.4H could better approximate the mean concentration at that height within the canopy. Consequently, concentration measurements at 2.8 m were excluded from all profiles prior to mass integration.

The temporal pattern of mass build-up were similar for $N_2O$ and $CO_2$ (Fig. 4). The increase in either $N_2O$ or $CO_2$ concentrations in the lowest 6.3 m corresponded with a decrease in wind speeds at 8 m (Fig. 2) as well as low $u_*$ and variance in w (Fig. 4). The mean gradient in $N_2O$ and $CO_2$ at this height during stable conditions and low turbulence was higher than that during higher turbulence, although the gradients varied widely (Table 3). If winds intermittently increase during the night, the concentration of both $N_2O$ and $CO_2$ decreased in the surface boundary layer, with an increase occurring after the winds decline again (Figs. 1, 3). This intermittent turbulence then mixed the heat and mass further into the developing nocturnal boundary layer. The accumulation of $CO_2$ and $N_2O$ in the lowest 8 m of the boundary layer might be expected to occur if the top of the layer exhibited minimal turbulence since the molecular diffusion of a gas is orders of magnitude smaller than the turbulent diffusion.

On average, the mean profiles of $CO_2$ and $N_2O$ concentrations during from 19 to 03 h LT showed nearly identical concentrations at 1 m and 5 m with decrease in concentration at 8 m (Fig. 5). The corresponding mean concentration profiles for the 03 to 07 h LT time window showed no change in concentration with height (Fig. 5). Conditions during the 1900 to 03 h LT period resulted in nearly identical mean wind speed profiles regardless of $u_*$ but substantially different temperature

profiles (Fig. 5). Temperature inversions above the canopy (2.8 m to 5 m agl) were evident between 19 and 03 h LT regardless of $u_*$ (Fig. 5). The temperature inversion was also evident between 03 and 07 h LT when $u_*$ was less than 0.05 ms$^{-1}$ (Fig. 5). This near-surface inversion was not evident at the top of the accumulation control volume (between 5 m and 8 m agl) where the wind shear was high.

## 3.2 Mass accumulations

Using the previously-defined top of the accumulation control volume, the accumulations of $N_2O$ and $CO_2$ were often evident during the night from 19 to 00 h LT with sunset approximately 21 h LT (Fig. 6). These mass accumulations corresponded with positive $z/\Lambda$ (locally stable conditions) and low $u_*$ (low turbulence). After quality assurance of the accumulated flux calculations, there were 97 90-min measurements of $N_2O$ nocturnal flux and 78 90-min measurements of $CO_2$ nocturnal flux with $u_*$ less than 0.05 ms$^{-1}$. Note that the mean gradients of both $N_2O$ and $CO_2$ were less for this set of measurements (Table 4) than for all measurement periods (Table 3).

Accumulated $N_2O$ flux during low turbulence and no measureable diffusive flux across the control volume top averaged 0.22 nmol m$^{-2}$s$^{-1}$ with a variability (standard deviation) greater than the mean (Table 4). The accumulated fluxes of $N_2O$ between 19 h LT and 03 h LT were relatively steady over the measurement period (Fig. 8). Accumulations within the control volume were greater (0.58 nmol m$^{-2}$s$^{-1}$) during the 22% of the measured flux periods when there was measureable diffusive flux out the top of the control volume (Table 4). When measureable, the diffusive flux of $N_2O$ was twice the accumulative flux (Table 4) resulting in a mean total measured $N_2O$ flux (accumulative + diffusive) of 0.53 nmol m$^{-2}$s$^{-1}$ (SD=0.25 nmol m$^{-2}$s$^{-1}$). This suggests that the ability to estimate diffusion across the upper boundary was limited by the $N_2O$ gradient. These fluxes were similar to median daily flux gradient-derived fluxes for maize grown over two years in a similar climate (Ontario, Canada) on imperfectly drained silt-loam soils with conventional tillage (0.5 nmol m$^{-2}$s$^{-1}$) but lower than that for no-till (Wagner-Riddle et al, 2007, Table 1).

The accumulated $CO_2$ fluxes between 19 h LT and 03 h LT generally decreased over time with values ranging from approximately 2.0 to 0.2 µmol m$^{-2}$s$^{-1}$ (Fig. 7). The mass accumulative flux during low turbulence averaged 0.40 µmol m$^{-2}$s$^{-1}$ with a variability less than the mean (Table 4). Measurable diffusive $CO_2$ flux out of the control volume, occurring 23% of the low turbulence $CO_2$ flux events, corresponded with only slightly lower accumulative fluxes (0.37 µmol m$^{-2}$s$^{-1}$; Table 4). This suggested that the limiting factor in estimating diffusion across the control volume was the turbulent exchange process not the concentration gradient. When measureable, the diffusive flux of was nine times the accumulative flux (Table 4), resulting in a mean total measured $CO_2$ flux (accumulative + diffusive) of 1.16 µmol m$^{-2}$s$^{-1}$. (SD=0.49 µmol m$^{-2}$s$^{-1}$). This flux was substantially lower than eddy-covariance-derived nocturnal mean flux over maize fields (10.8 to 30.0 µmol m$^{-2}$s$^{-1}$) in a similar climate (Ontario, Canada) during the same period in the growing season but under more turbulent winds: mean wind speeds of at least 1.5 ms$^{-1}$ and $u_*$ between 0.075 and 0.1 ms$^{-1}$ (Pattey et al, 2002) compared to mean wind speeds of 1.1 ms$^{-1}$ and $u_*$ of 0.02 ms$^{-1}$.

Greater turbulence (higher $u_*$ at 8 m) did not affect the accumulative $N_2O$ flux in the control volume if no diffusion was measurable but did reduce the flux when there was measureable diffusion (Table 4). Greater turbulence reduced the accumulative $CO_2$ flux whether or not there was measureable diffusion (Table 4). The greater turbulence corresponded with a decrease in the mean $N_2O$ gradient and an increase in the $CO_2$ gradient at the top of the control volume and increased diffusive flux out of the control volume (Table 4). The upper transport 'cap' to the mass accumulation control volume was on average stronger for the low turbulence condition than the higher turbulence condition (based on $\sigma_w$ and $\sigma_w/u_*$; Table 2) and the eddy diffusivities were lower (Table 3). The effectiveness of this 'cap', separating the developing nocturnal boundary layer above from the surface boundary layer below, had a larger effect on the mass accumulation of $CO_2$ than $N_2O$ and a greater effect on the diffusive flux of $N_2O$ than $CO_2$ (Table 4). This might be expected if the local $CO_2$ flux was more similar to the more distant surroundings (more homogeneous) than the $N_2O$ flux. It is important however to note that the high variability in $CO_2$ and $N_2O$ fluxes under low turbulence resulted in mean accumulative fluxes with or without measureable diffusive flux that was not statistically different (Student t-test; p=0.05) (Table 4).

Eddy diffusivities were comparable to and exhibited the same relationship to $u_*$ and $z/\Lambda$ for positive $z/\Lambda$ as those reported for $N_2O$ and $NH_3$ in Schaefer et al. (2012). The mean eddy diffusivities were more than an order of magnitude higher for conditions with $u_* > 0.05$ ms$^{-1}$ than $u_* < 0.05$ ms$^{-1}$ (Table 3). Clearly the $u_*$ threshold of 0.05 ms$^{-1}$ still allowed for weak turbulent diffusion of both $N_2O$ and $CO_2$ out of the near-surface control volume and into the nocturnal boundary layer in 22% and 23% (respectively) of the flux events (Table 4). The general relatively high diffusive versus accumulative flux (Table 4) during low turbulence conditions however may also be a result of a combination of non-stationarity of the flow and/or anemometer tilt. Assuming stationary flow and no anemometer tilt, the approximation of the eddy diffusivities of $N_2O$ and $CO_2$ by substituting molecular Schmidt numbers for turbulent Schmidt numbers likely contributed to underestimated flux values since these Schmidt numbers were higher than the generalized turbulent Schmidt number of Flesch et al (2002).

### 3.3 Soil chamber fluxes

The soil chamber $CO_2$ and $N_2O$ flux measurements, made at various hours of the day during the measurement period, also showed a decreasing flux over the period (Figs. 7, 8). $CO_2$ flux in the 200Sp treatment, where the profile measurements were made, ranged from 0.1 µmol m$^{-2}$s$^{-1}$ to 2.1 µmol m$^{-2}$s$^{-1}$ and averaged 0.9 µmol m$^{-2}$s$^{-1}$. These chamber measurements had a mean signal to noise ratio of 250. These fluxes are similar to soil+root respiration fluxes reported in the literature for maize fields (Table 1). The region of the south field in which no N was applied during the past year (Fig. 1) had a mean $CO_2$ emission of 0.5 µmol m$^{-2}$s$^{-1}$, averaging 50% of the mean field emissions under various N treatments and similar to that reported for soil+root respiration of soybean in the literature (Table 1). Although most measurements were made at 23 h LT, some of the variability in chamber measurements was a result of the time of measurement. The four-day study of diurnal variation in mean hourly $CO_2$ emissions ranged from 1.04 µmol m$^{-2}$s$^{-1}$ to 1.48 µmol m$^{-2}$s$^{-1}$ with the highest emissions at 18 h LT with a ratio of midnight to noon LT emissions of 1.2.

Nitrous oxide fluxes in the 200Sp treatment field ranged from 0.3 nmol m$^{-2}$s$^{-1}$ to 2.2 nmol m$^{-2}$s$^{-1}$ averaging 1.1 nmol m$^{-2}$s$^{-1}$. These fluxes were lower than commonly reported in the literature for maize but similar to that of soybeans (Table 1). This may be due to the negligible amount of the applied nitrogen available for denitrification and nitrification in the maize field. These chamber N$_2$O measurements thus had a mean signal to noise ratio of 1.7. The fields on which no N was applied during the year had a mean emission of 0.59 nmol m$^{-2}$s$^{-1}$; 54% of the mean fertilized field emissions and equal to the Chamber method MDL. As with the CO$_2$ flux measurements, some of the variability in chamber measurements was a result of the time of measurement. The four-day study of diurnal variation in mean hourly N$_2$O emissions ranged from 0.96 nmol m$^{-2}$s$^{-1}$ to 1.40 nmol m$^{-2}$s$^{-1}$ with the highest emissions at 18 h LT with a ratio of midnight to noon LT emissions of 0.93.

### 3.4 Comparative fluxes

As with the comparison of CO$_2$ fluxes determined by eddy covariance and boundary-layer mass balance (Eugster and Siegrist, 2000), the fluxes determined by chamber and mass accumulation are local and 'regional' fluxes respectively. The CO$_2$ flux measurements based on mass accumulation within the control volume but not diffusion across the control volume top were generally lower than the chamber measurements with the exception of a few outlier high mass accumulation values (Fig. 7). Inclusion of measureable diffusive flux to the accumulative flux resulted in total flux estimates more similar to soil chamber measurements. Average mean daily CO$_2$ flux estimates for two of the three measurement time periods indicated the total mass accumulation method flux was between 0.9 and 1.3 of that determined by the chamber method (Table 5). Higher accumulation flux over the chamber flux was expected due to the chamber flux method measured only root and soil respiration while the mass accumulation flux method measured the respiration of the soil, roots, stalks and leaves. This can result in a large difference in flux: Parkin et al (2005) measured soil and root respiration with chambers and whole canopy respiration by eddy covariance and found that the soil respiration was approximately 50% of the total measured CO$_2$ flux. Given the variability in daily flux estimates within each period, the fluxes determined by chamber and mass accumulation methods were not significantly different (Table 5).

The N$_2$O flux measurements based on mass accumulation under low turbulence and stable conditions were generally much lower than those measured using the chambers on the same day (Fig. 8). Inclusion of measureable diffusive fluxes in the flux estimates over three measurement time periods showed that the accumulation method estimated mean daily fluxes only 60% to 80% of the soil chambers (Table 5). Again, given the variability in mean daily flux estimates within each time period, the fluxes determined by the chamber and mass accumulation methods were not significantly different (Table 5).

Differences between the accumulation flux versus chamber flux measurements were likely in part due to the advection of gas emitted from surrounding fields. The accumulated mass of CO$_2$ and N$_2$O have contributions from local soils sources as well as mass advection from more distant sources due to the meandering nature of the air flow during the stable nocturnal conditions (Eugister and Siegrist, 2000). Unfortunately, the analytical approaches to defining the flux footprint do not apply to the stable nocturnal conditions in which the accumulations occur (z/$\Lambda$>+1, u$_*$< 0.05 ms$^{-1}$; Vesala et al, 2007), although they are believed

to be in the order of ten kilometers (eg. Chambers et al, 2011). At scales of kilometres (10 km$^2$ area), the land use was crop agriculture; dominated by nearly equal soybean and maize production (46% and 47% respectively with an addition 2% in grass in the (Table 1). Within the nearest square kilometre around the research site, maize production dominated the land use (Table 1).

The $CO_2$ flux of the un-fertilized fields were similar to those of the fertilized fields (Fig. 7). The measured fluxes were substantially lower than those for other maize fields as well as grass and soybean fields reported in the literature (Table 1). If anything, it is reasonable to assume that the advected, regionally-emitted $CO_2$ from surrounding soybean and maize production would have increased the accumulation flux estimates. However the relatively low accumulation fluxes suggest that advection did not substantially contribute to the measured mass accumulation. The measured chamber $N_2O$ flux from un-fertilized fields

of maize was typically lower than fertilized maize fields and closer to the flux measured by the accumulation method (Fig. 8). Since roughly one-half the surrounding area was in soybean production (Table 1), it is reasonable to assume horizontal advection of air with higher $N_2O$ concentration from nearby grass and soybean canopies could have potentially affected the $N_2O$ profile. However, literature values for fluxes from surrounding grassy areas and soybean fields (Table 1) are generally similar to the flux measured by the accumulation method in a fertilized maize field (Table 5). Consequently there is little

evidence to support the supposition that advection contributed significantly to the accumulated mass.

The general underestimate of $CO_2$ and $N_2O$ fluxes using the mass accumulation method may also be a result of using two small of an accumulation volume. The 'cap' of the volume was arbitrarily set at the geometric mean between the upper two measurement heights. An objective measure of the 'cap' height is needed. Given the significantly greater flux associated with diffusion out the top of the accumulation control volume relative to the computed accumulated flux within the control volume

(Table 4), the accumulation control volume was likely too shallow.

## 4 Conclusions

Nocturnal $CO_2$ and $N_2O$ emissions from the soil surface were determined by measuring the accumulation of mass within a mixing-limited surface boundary layer control volume and the diffusion of mass out the top of the control volume. The magnitude of the accumulations influenced the ability for the accumulation method to be effective at estimating nocturnal flux:

$CO_2$ flux determined by the accumulation method were comparable to those measured using the chamber method while that for $N_2O$ were below that measured using the chamber method. For the $N_2O$ flux, there is no known canopy flux of $N_2O$ and consequently the chamber method and accumulation method should have been comparable. Measurement errors associated with a limited vertical dimension to the control volume, non-stationarity of low turbulent flow in the stable nocturnal surface boundary layer, and estimating the Schmidt number for the diffusive flux component likely contributed to the differences

between the accumulation and chamber flux methods. Advection during the stable nocturnal conditions did not appear to contribute to the measured profiles and the subsequent estimate of $N_2O$ flux or $CO_2$ flux. Additional work is needed to evaluate the use of the accumulation method for $N_2O$ fluxes for accumulations within a larger vertical domain to the control volume

and more homogeneous regional land use in conjunction with using chamber methods with a lower MDL (higher signal to noise ratio).

## Author Contribution

R. Grant designed, conducted and analysed the mass accumulation experiment while R. Omonode conducted the chamber gas
flux measurements. R. Grant prepared the manuscript with contributions from R. Omonode.

## Competing interests

The authors declare that they have no conflict of interest.

## Acknowledgements

The authors appreciate the field technical assistance of Cheng Hsien Lin and Austin Pearson.

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

**Table 1: 2015 Land use around the research site and literature-reported CO$_2$ and N$_2$O fluxes for each land use.**

| Land use | 1 km$^2$ area (%)[1] | 10 km$^2$ area (%)[1] | CO$_2$ respiration ($\mu$mol m$^{-2}$ s$^{-1}$) | Source | N$_2$O emissions (nmol N$_2$O m$^{-2}$ s$^{-1}$) | Source |
|---|---|---|---|---|---|---|
| Maize production | 83 | 47 | Soil/root: 0.9-1.8<br><br>Canopy:11.7-15.8 | Omonode et al, 2007<br>Pattey et al, 2002 | 0- 2.1<br>0.5-2.3<br>0.2-0.5 | Eichner, 1990<br>Parkin and Kaspar, 2006<br>Wagner-Riddle et al, 2007 |
| Soybean production | 15 | 46 | Soil/root: 2.9<br><br>Canopy: 3.6,4.3<br>Soil/root: 0.41,0.49<br>Canopy: 3.8<br>Canopy:17.5 | Raich & Tufekcoglu, 1999;<br>DeCosta, et al, 1986<br><br>Parkin et al, 2005<br><br>Pattey et al, 2002 | 0.3-1.2<br><br>1.7-2.0 | Bemner et al, 1980<br><br>Parkin and Kaspar, 2006 |
| Grass | 2 | 2 | Canopy: 3.5 | Tufekcoglu,et al 2001 | 0.2<br>0.3 | Eichner, 1990<br>Mosier et al, 1991 |
| Deciduous Forest | 0 | 1 | Soil/root: 2.2,2.5<br><br>Canopy: 5.2 | Raich &Tufekcoglu, 1999<br>Lee at al, 1996 | <0.3-0.6, 1.2 | Bowden et al, 2000;<br>Goodroad & Keeney, 1984 |
| Bare ground | 0 | <1 | Soil: 0.06,0.06,0.06 | DeCosta, et al, 1986 | 1.4-2.0 (fertilized) | Bremner et al, 1981 |
| Alfalfa | 0 | <1 | Canopy: 1.7<br>Soil/root: 1.1 | DeCosta, et al, 1986 | 1.7-4.3 | Duxbury and Bouldin, 1982 |

1: Land use during the 2015 growing season assessed using CropScape Cropland Data Layer (USDA,2017).

**Table 2: Wind conditions over the maize canopy. Statistics based on 30-min averaging period of 10Hz 3D sonic anemometer measurements at indicated heights over the entire study period.**

| Time interval (LT) | Flow condition at 8 m | Statistic | 8 m | | 5 m | | | 8 m | | |
|---|---|---|---|---|---|---|---|---|---|---|
| | | | $U^{\dagger}$ (ms$^{-1}$) | $z/\Lambda^{\ddagger}$ | Friction velocity-$u_*$ (ms$^{-1}$) | Standard deviation of vertical velocity– $\sigma_w$ (ms$^{-1}$) | $\sigma_w/u_*$ | Friction velocity-$u_*$ (ms$^{-1}$) | Standard deviation of vertical velocity–$\sigma_w$ (ms$^{-1}$) | $\sigma_w/u_*$ |
| 1900-0300 | Low turbulence $u_*\leq0.05$ ms$^{-1}$ $n^{+}=290$ | Mean | 1.05 | 16.05 | 0.04 | 0.003 | 0.066 | 0.02 | 0.002 | 0.080 |
| | | Standard deviation | 0.45 | 0.80 | 0.02 | 0.003 | 0.152 | 0.01 | 0.002 | 0.176 |
| | Turbulent $u_*>0.05$ ms$^{-1}$ $n=314$ | Mean | 2.17 | 0.10 | 0.21 | 0.089 | 0.421 | 0.19 | 0.083 | 0.435 |
| | | Standard deviation | 0.94 | 0.04 | 0.14 | 0.067 | 0.488 | 0.13 | 0.104 | 0.800 |
| 0300-0700 | Low turbulence $u_*\leq0.05$ ms$^{-1}$ $n=157$ | Mean | 0.98 | -3.43 | 0.04 | 0.003 | 0.072 | 0.03 | 0.002 | 0.086 |
| | | Standard deviation | 0.44 | 0.32 | 0.02 | 0.004 | 0.204 | 0.01 | 0.004 | 0.322 |
| | Turbulent $u_*>0.05$ ms$^{-1}$ $n=923$ | Mean | 2.80 | -1.33 | 0.36 | 0.212 | 0.593 | 0.33 | 0.200 | 0.605 |
| | | Standard deviation | 1.45 | 0.00 | 0.17 | 0.188 | 1.090 | 0.17 | 0.171 | 1.021 |

$\dagger$: U=wind speed

$\ddagger$: $\Lambda$= Local Obukhov length

+: n= number of 30-min measurements

**Table 3: Characteristics of the nocturnal boundary layer at the top of the accumulation control volume with stable conditions (positive local Obukhov length) at 8m. Statistics based on 30-min averaging periods.**

| Time interval (LT) | Flow condition at 8 m agl | Statistic | 6.3 m agl | | | | |
|---|---|---|---|---|---|---|---|
| | | | $\Delta T_s^{\dagger}/\Delta z$ (ºC m⁻¹) | $\Delta N_2O/\Delta z$ (μmol m⁻⁴) | $\Delta CO_2/\Delta z$ (mmol m⁻⁴) | $K^{\ddagger}_{N2O}$ (m² s⁻¹) | $K_{CO2}$ (m² s⁻¹) |
| 1900-0300 | Low turbulence $u_*^{+}<=0.05$ ms⁻¹ | Mean | -0.008 | 0.08 | 1.11 | 0.008 | 0.008 |
| | | Standard deviation | 0.033 | 0.14 | 1.23 | 0.024 | 0.022 |
| | Turbulent $u_*>0.05$ ms⁻¹ | Mean | 0.148 | 0.00 | 0.21 | 0.233 | 0.221 |
| | | Standard deviation | 0.025 | 0.09 | 0.38 | 0.229 | 0.216 |
| 0300-0700 | Low turbulence $u_*<=0.05$ ms⁻¹ | Mean | 0.005 | 0.06 | 0.82 | 0.010 | 0.009 |
| | | Standard deviation | 0.053 | 0.09 | 0.96 | 0.111 | 0.105 |
| | Turbulent $u_*>0.05$ ms⁻¹ | Mean | 0.270 | 0.00 | 0.02 | 0.601 | 0.568 |
| | | Standard deviation | 0.035 | 0.19 | 0.18 | 0.307 | 0.290 |

+: $u_*$ =friction velocity

†: $T_s$ =sonic temperature

‡: K =diffusion coefficient

n

**Table 4: Flux of $N_2O$ and $CO_2$ across the top of the accumulation control volume during stable (positive local Obukhov length) nocturnal conditions between 19 and 07 h LT. Accumulation flux based on 90-min mass accumulations. Diffusive flux based on average of three ½ hour gradients.**

| Flow condition at 8 m | Statistic | Gradient at top of control volume (6.3 m agl) $N_2O$ ($\mu$mol m$^{-4}$) | $CO_2$ (mmol m$^{-4}$) | $N_2O$ accumulation flux (nmol m$^{-2}$s$^{-1}$) without measurable diffusion | with measurable diffusion | $N_2O$ diffusive flux (nmol m$^{-2}$s$^{-1}$) | $CO_2$ accumulation flux ($\mu$mol m$^{-2}$s$^{-1}$) without measurable diffusion | with measurable diffusion | $CO_2$ diffusive flux ($\mu$mol m$^{-2}$s$^{-1}$) |
|---|---|---|---|---|---|---|---|---|---|
| Low turbulence $u_*{}^+ <= 0.05$ ms$^{-1}$ | Mean | 0.04 | 0.43 | 0.22 | 0.58 | 1.06 | 0.40 | 0.37 | 3.33 |
| | SD[‡] | 0.06 | 0.48 | 0.17 | 0.62 | 1.37 | 0.29 | 0.32 | 3.58 |
| | n[†] | 89 | 67 | 76 | 21 | 21 | 60 | 18 | 18 |
| Turbulent $u_* > 0.05$ ms$^{-1}$ | Mean | 0.02 | 0.64 | 0.19 | 0.49 | 3.23 | 0.11 | 0.13 | 5.87 |
| | SD | 0.04 | 0.64 | 0.18 | 0.82 | 1.92 | 0.11 | 0.11 | 2.48 |
| | n | 59 | 59 | 58 | 5 | 5 | 65 | 26 | 26 |

+: $u_*$=friction velocity

‡: SD=standard deviation

†: n= number of 90-min values

**Table 5: Comparative mean daily fluxes of $N_2O$ and $CO_2$ across three similar flux periods.**

| Measurement period (DD/MM/YY) | Total mass accumulation flux (including accumulative and diffusive flux when measureable) | | | Chamber flux in Sp200 treatment field | | | Ratio |
|---|---|---|---|---|---|---|---|
| | n* | Mean | Standard deviation | n* | Mean | Standard deviation | Mass accumulation/ Chamber |
| $CO_2$ | (#) | ($\mu$mol m$^{-2}$s$^{-1}$) | ($\mu$mol m$^{-2}$s$^{-1}$) | (#) | ($\mu$mol m$^{-2}$s$^{-1}$) | ($\mu$mol m$^{-2}$s$^{-1}$) | |
| 22/07/15-31/07/15 | - | - | - | 2 | 1.60 | 0.37 | - |
| 01/08/15-22/08/15 | 12 | 0.76 | 0.60 | 11 | 0.83 | 0.44 | 0.9 |
| 23/08/14-2/09/15 | 4 | 0.21 | 0.08 | 2 | 0.16 | 0.10 | 1.3 |
| $N_2O$ | (#) | (nmol m$^{-2}$s$^{-1}$) | (nmol m$^{-2}$s$^{-1}$) | (#) | (nmol m$^{-2}$s$^{-1}$) | (nmol m$^{-2}$s$^{-1}$) | |
| 23/07/15-31/07/15 | 5 | 1.20 | 0.83 | 2 | 1.93 | 0.34 | 0.6 |
| 01/08/15-22/08/15 | 14 | 0.76 | 0.18 | 11 | 1.00 | 0.35 | 0.8 |
| 23/08/14-10/09/15 | 7 | 0.25 | 0.15 | 2 | 0.40 | 0.22 | 0.6 |

*= number of days with valid measurements

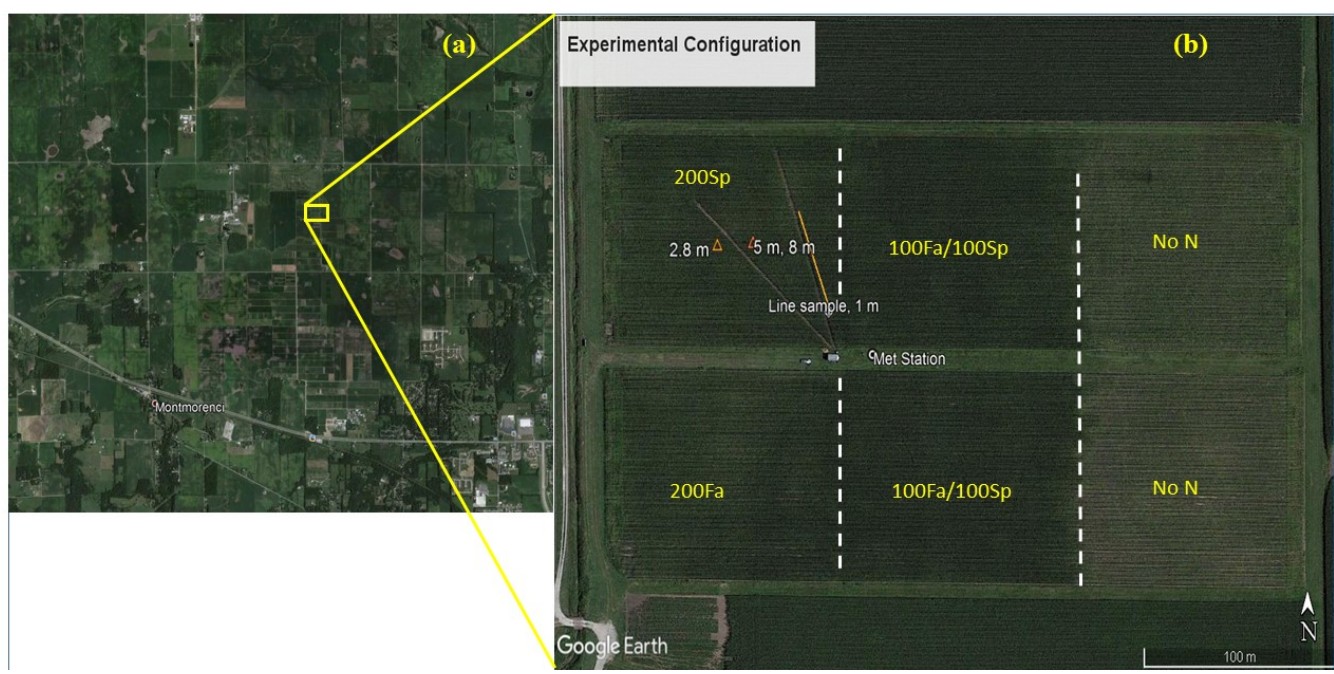

Figure 1: Experimental domain: GoogleEarth® images from August 2015 showing the homogeneous agricultural land use across the region surrounding the experimental field (Montmorenci, Indiana, USA; 40.495º latitude and -86.994º longitude: panel a) and the configuration of measurements in the experimental field (panel b). Fertilizer treatments (kgN Ha$^{-1}$/Season) in the north field

10    and south field (200Sp, 200Fa, 100Sp/100Fa, and No N) as well as the locations and heights of the sonic anemometers and inlets (open triangles), integrated line sample (open diamond and orange line), and meteorological station (open circle) are indicated in panel b. Note scale in lower right corner of panel b.

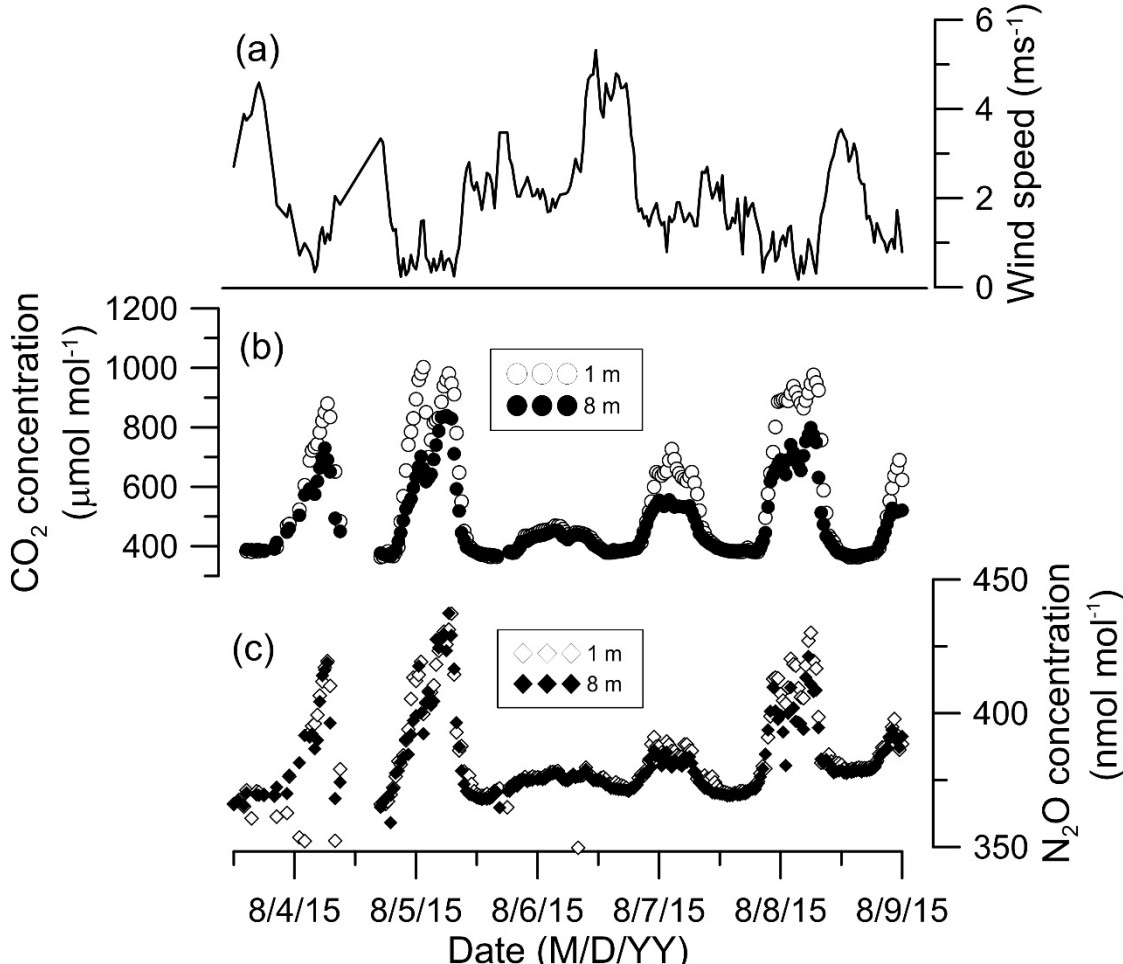

**Figure 2: Changes in $CO_2$ and $N_2O$ concentrations at the bottom and top of the measured control volume relative to wind speed at 8 m. The wind speed at 8m (panel a, right ordinate), the $CO_2$ concentrations at 1 m and 8 m (panel b, left ordinate) and the $N_2O$ concentrations at 1 m and 8 m (panel c, right ordinate) are indicated for a five-day period. Dates on the abscissa are indicated at the beginning of the indicated day (midnight). Note the increase in wind speed during the 8/5/15 night corresponds with a decrease in both the 1 m and 8 m concentrations of both $CO_2$ and $N_2O$. Date/time is local time.**

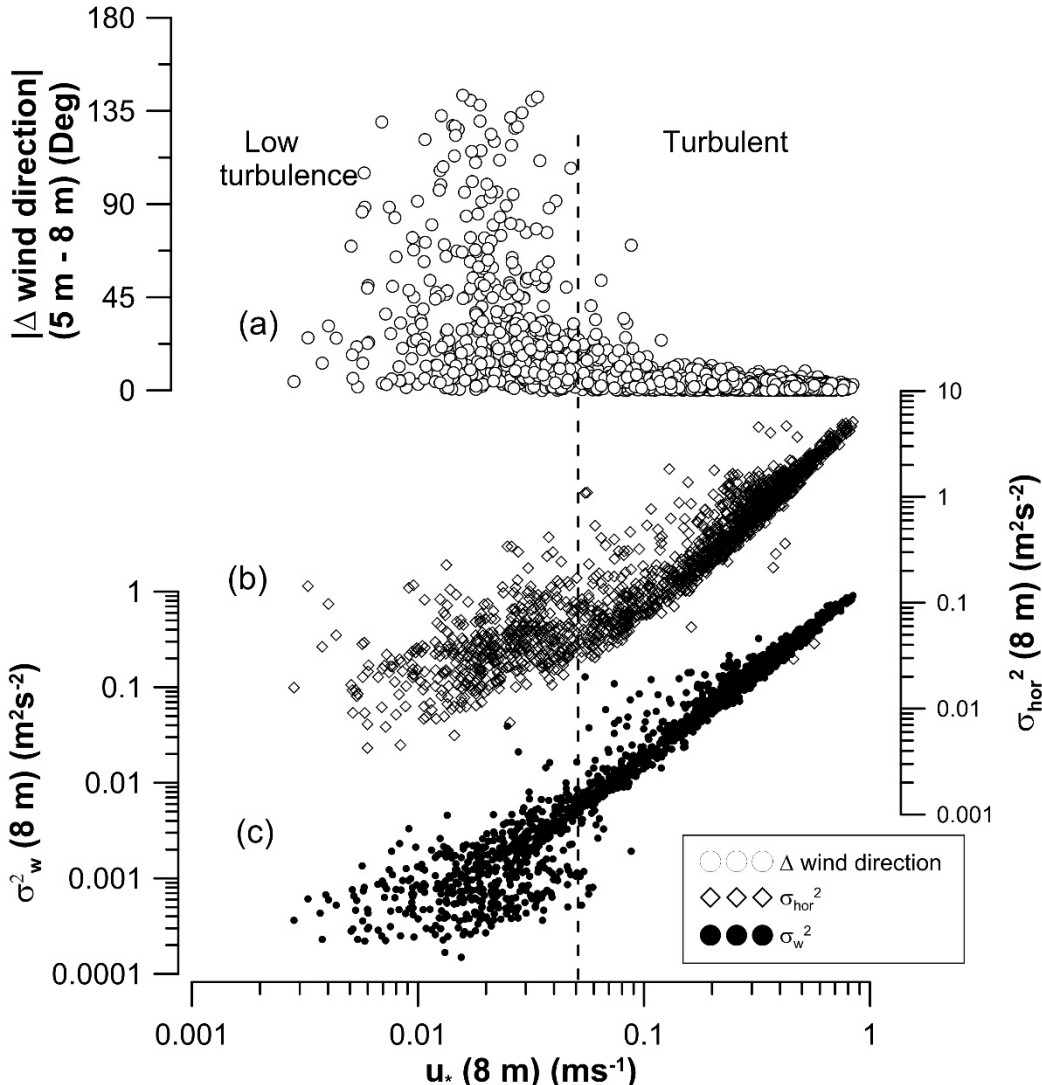

**Figure 3: Wind conditions in the near-surface layer over the entire study period. The relationship between absolute value of change in wind direction (panel a with ordinate axis to left), horizontal wind velocity variance ($\sigma_{hor}^2$; panel b with ordinate axis to left) and the vertical wind velocity variance ($\sigma_w^2$; panel c with ordinate axis to right) with friction velocity ($u_*$) is indicated. The dashed line demarcates the separation of 'low turbulence' and 'turbulent' classifications for wind conditions. Note that the demarcation between 'low turbulence' and 'turbulent' flow corresponds with a $\sigma_w^2$ threshold of 0.01 m²s⁻².**

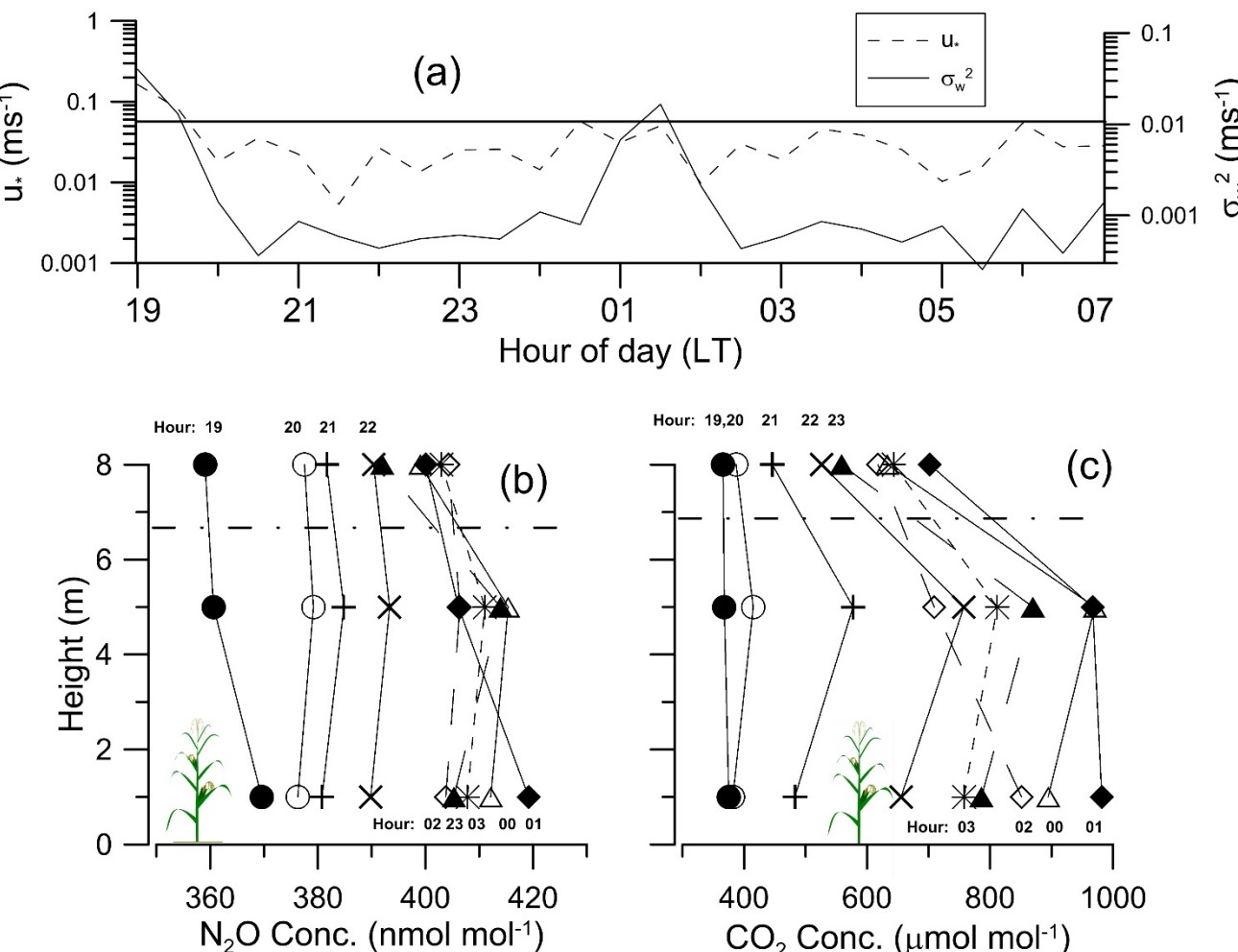

**Figure 4: Near-surface atmospheric conditions during the night of 5 August, 2015. The friction velocity ($u_*$, left ordinate) and vertical wind velocity variance ($\sigma_w^2$, right ordinate) at 8 m are indicated from 19 to 07 h LT in panel a. The solid line (panel a) indicates the upper thresholds for the 'low turbulence' classification. Labelled profiles (h LT) of $N_2O$ and $CO_2$ concentrations every hour from 19 h LT until 03 h LT are indicated with differing symbols and lines in panels b and c. Note the 01-02 h LT burst of vertical wind variance (panel a) corresponds with losses in $N_2O$ (panel b) and $CO_2$ (panel c). Sunrise and sunset times were approximately 07 h and 21 h LT.**

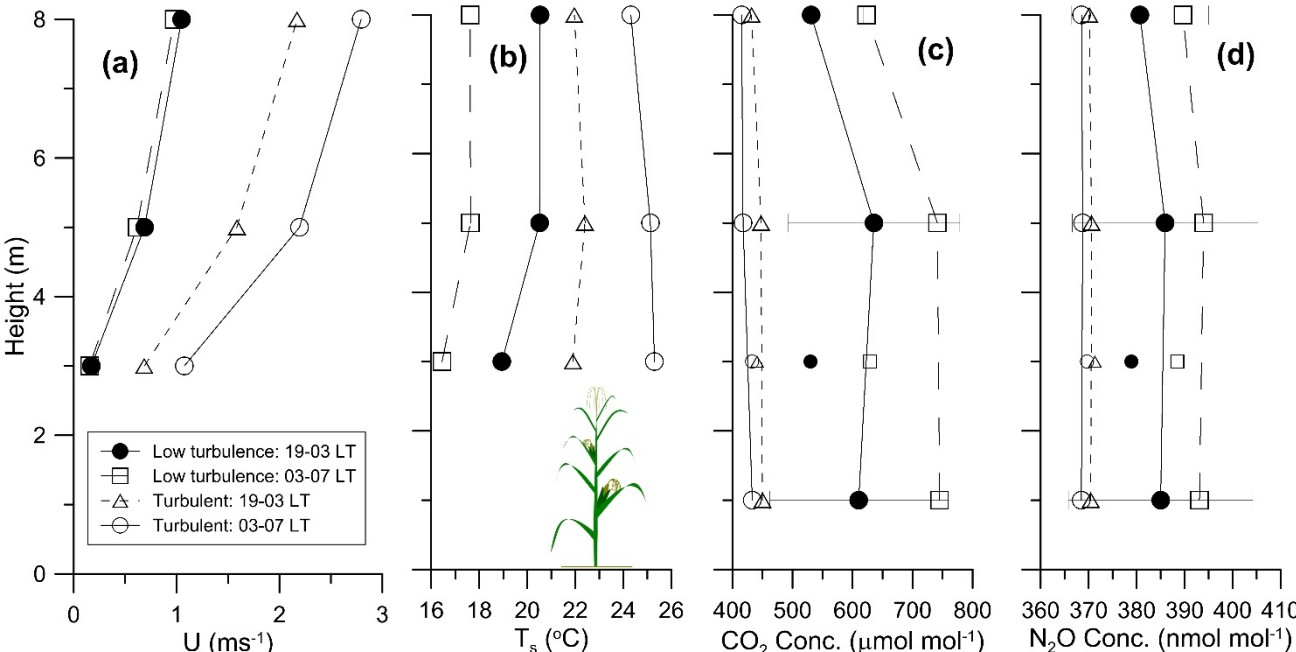

**Figure 5: Mean profiles of wind speed, sonic temperature, and concentrations of $CO_2$ and $N_2O$ under different friction velocity and time domain classes for the entire study period. The mean wind speed (U, panel a), sonic temperature ($T_s$; panel b), and concentration profiles of $CO_2$ (panel c) and $N_2O$ (panel d) when the air at 8 m had low turbulence ($u_* < 0.05$ ms$^{-1}$) or turbulent ($u_* >= 0.05$ ms$^{-1}$) between 19 and 03 h LT and 03 and 07 h LT are indicated. Canopy height was 2.8 m. Smaller symbols not connected with lines represent concentration measurements excluded from mass accumulations due to their close proximity to the canopy top.**

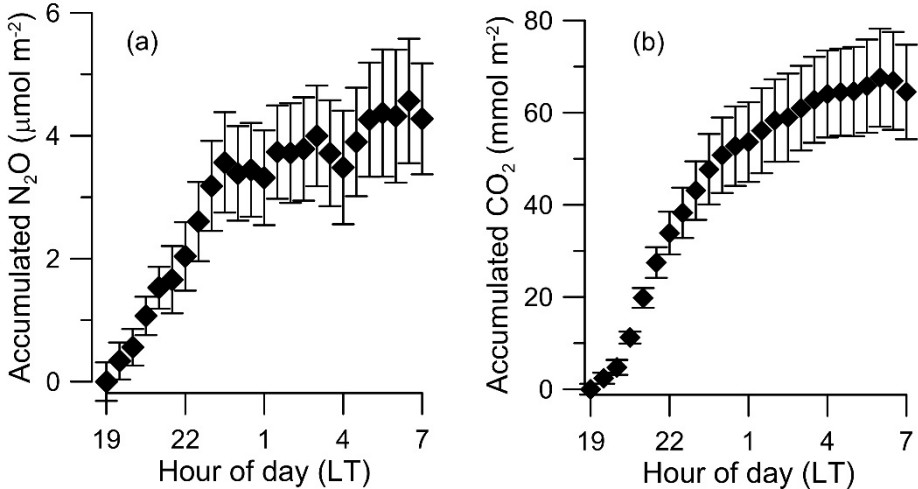

**Figure 6: Accumulation of CO₂ and N₂O within the lowest 6.3m of the boundary layer during the night throughout the study period. The mean accumulations of N₂O (panel a) and CO₂ (panel b) are indicated with vertical error bars indicating the standard error of the mean of each 30-min mean accumulation. Sunrise was approximately 06 to 07 h LT.**

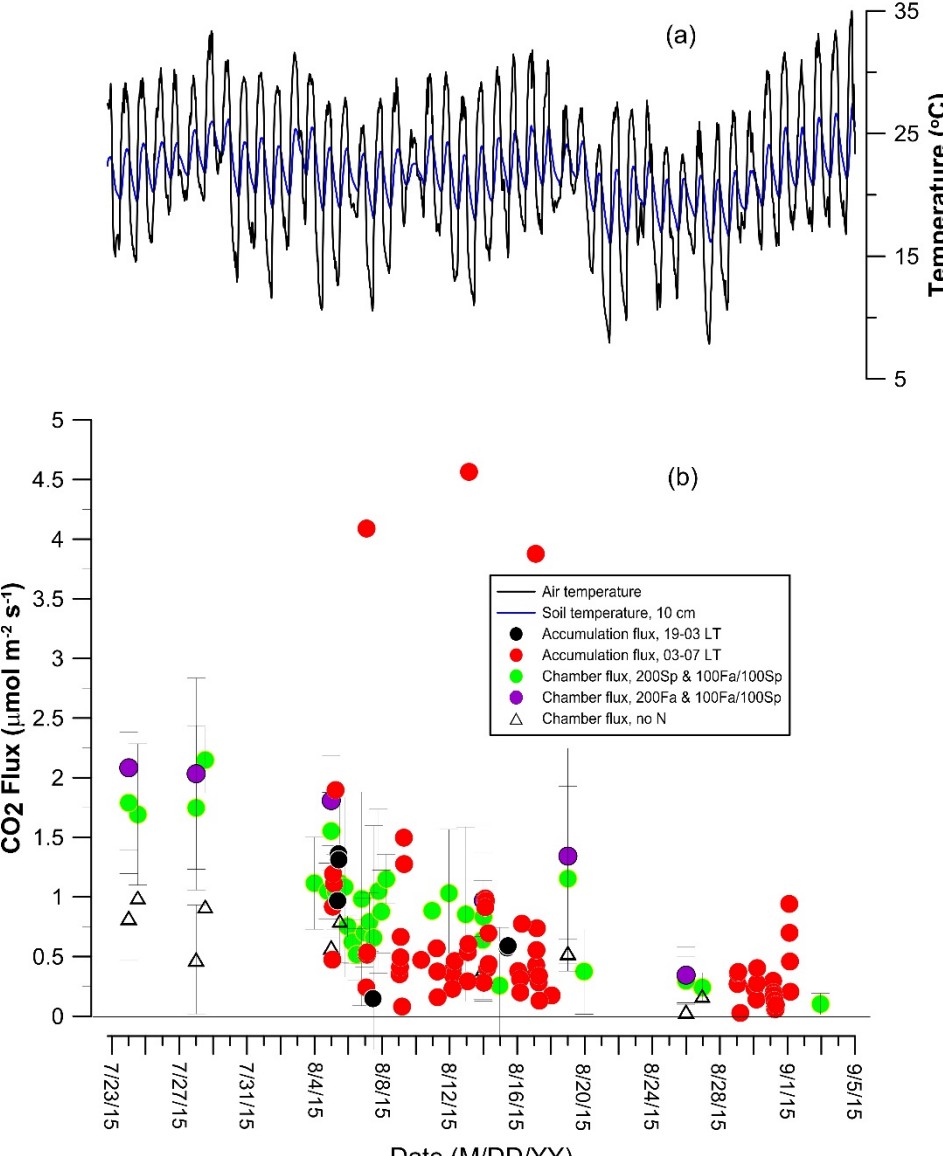

**Figure 7: Temperatures and CO₂ flux based on accumulation and chamber methods. Diurnal variation in air (solid black line) and 10 cm soil at 10 cm depth (dashed blue line) during the period are indicated in panel a. Total fluxes (accumulative within and diffusive out the top of the control volume) under stable, low turbulence conditions and soil+root fluxes calculated using the chamber method are indicated in panel b (ordinate axis with differing units to left and right). The standard deviation of the three chamber flux measurements in each field are indicated by the vertical bars.**

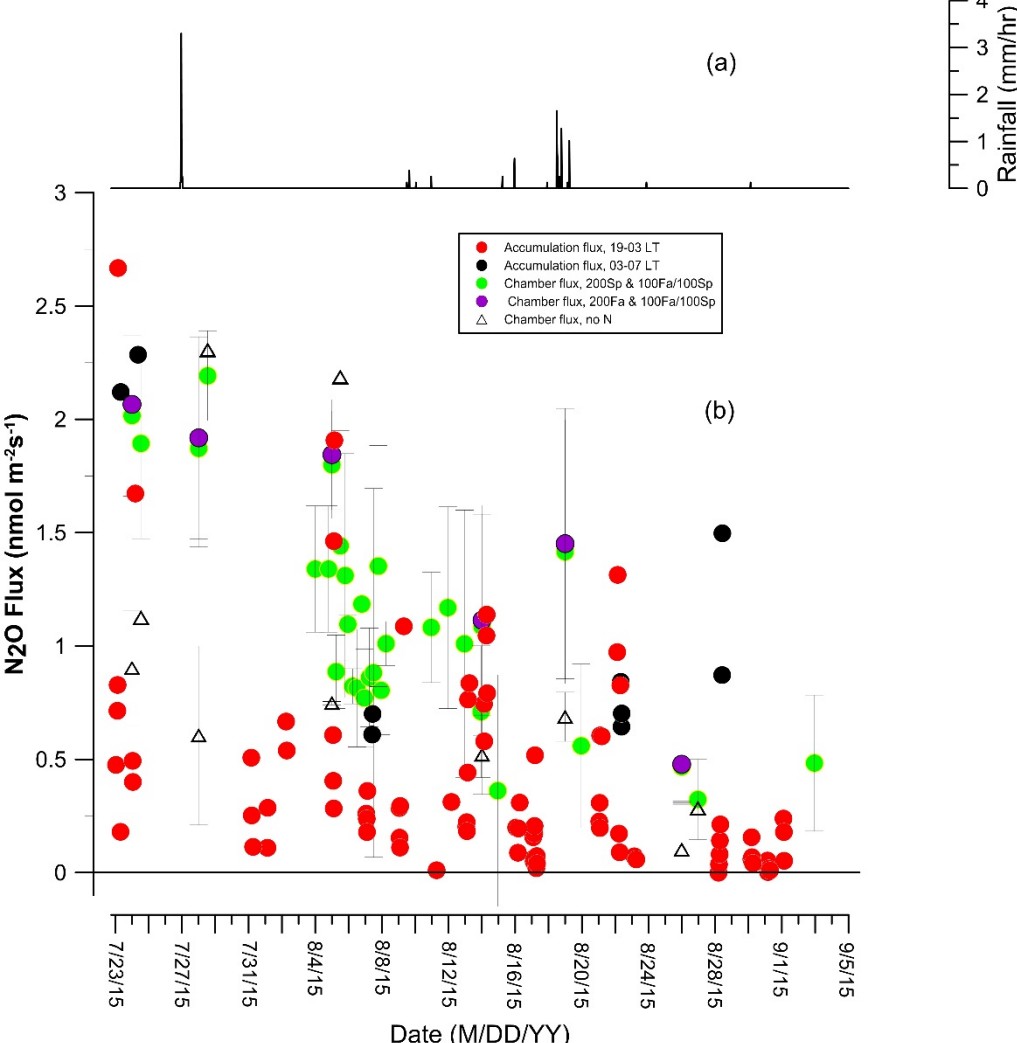

**Figure 8: Precipitation and N₂O flux based on accumulation and chamber methods. Precipitation is indicated in panel a. Total fluxes (accumulative within and diffusive out the top of the control volume) under stable, low turbulence conditions and soil+root fluxes calculated using the chamber method are indicated in panel b (ordinate axis with differing units to left and right). The standard deviation of the three chamber flux measurements in each field are indicated by the vertical bars.**