# Peer review of "Estimation of nocturnal CO2 and N2O soil emissions from changes in surface boundary layer mass storage"

_Atmospheric Measurement Techniques, 2017_

## Referee Comment (RC1) · Anonymous Referee #1 · 15 Sep 2017

Observations of turbulence and gas concentrations over a flat, agricultural terrain are analysed in this manuscript and show that gas accumulation in the nocturnal boundary layer can provide reasonable estimates of CO2 and N2O emissions. The site, meteorological conditions, and measurement installations were ideal for this approach. The results clearly show potential and limitations of the technique. In this sense, the study makes a useful contribution to the journal.

The only major addition I would like to propose is a broader discussion of the technique in the context of other techniques used to estimate gas exchange between land and atmosphere. In particular, I would like to see a comparison with the eddy covariance

and the radon mass balance techniques (e.g. Biraud et al., 2002, Tellus, 54B, 41-60) in terms of their precision and the scale of the observed 'footprint'.

Minor issues

Title: instead "...using changes..." perhaps "...from changes..."?

Page 2, line 12: "...mass accumulations are reported for CO2, CH4, N2O, and H2..." Since H2 is consumed by soil microorganisms, I would expect H2 concentrations to decrease in the nocturnal boundary layer, not to accumulate.

Methods: Please show coordinates of the experimental field, or at least tell the reader in which country, near which town, it is located.

Page 3, line 30: "measured", not "measure"

Precision of reported fluxes, e.g., page 7, line 15, and Table 4: How meaningful is it to report the value of a mean flux to the second digit after the decimal point, when the standard deviation is larger than the mean itself?

Mass accumulations, first paragraph: Were the comparable fluxes cited here done in a similar climatic region, with similar land management (e.g. N fertilisation)?

Page 8, line 29: The first sentence in this line states a trivial fact and can be deleted.

Page 9, Discussion of lower N2O accumulation compared to chamber fluxes: Another possible explanation is that chamber fluxes were measured during the day, when soils tend to be warmer than during the night. Other parameters being equal, N2O flux from soil increases substantially with soil temperature.

———————————

---

## Referee Comment (RC2) · Anonymous Referee #2 · 12 Oct 2017

**General Comments**

This manuscript describes measurements on carbon dioxide and nitrous oxide concentration increases in the nocturnal stable surface layer to arrive at fluxes. The technique is not new, but the manuscript provides additional data to the scientific community.

Overall, the methods are sound and the structure of the paper is appropriate. The study covers a period of low N2O fluxes, which creates some additional challenges for the measurements. Below are several specific comments that the author should address.

Specific Comments:

[Figure]

1. Page 1, line 24. The use of the concentration change within the stable surface layer is also a "micrometeorological" technique. Your method is one of the micromet tools available.

2. Page 1, line 26. The community usually uses "eddy covariance" instead of "eddy correlation".

3. Page 2, line 21. I couldn't find that SBL was defined.

4. Page 2, line 22. Molecular diffusion rates are closer to 10ˆ-5.

5. Page 2, line 22. Qualify that you mean typical turbulent diffusion coefficients during daytime. We can argue a wide range before we get to molecular diffusivity at night.

6. Page 3, line 26. I think you mean that the N2O MDL is 0.3 nL/L, not ul/L?

7. Page 3, line 28. The manual for this instrument suggests better than 1 uL/L; is your value related to precision or accuracy?

8. Page 4, line 5. The van de Wiel reference is quite recent, whereas similarity theory has been developed much earlier. Please give original references.

9. Check typographical (spelling) errors: e.g., Page 4 line 19 and line 26.

10. Page 4, line 27. Why were the chamber measurements made during the day? Can you give the audience an indication about how the chamber measurements would cycle diurnally? Recall that your comparison is with the night.

11. Page 5, line 6. You use 30-minute chamber measurements for a relatively short period on each day. It would seem more reasonable to report the measurements on a reasonable time unit; typically umol/m2/s is used. It is misleading to scale this to units of "per day" with such a small, biased sample.

12. Page 6, line 1. Be consistent; use friction velocity instead of shear velocity here. Also, in several places, variance is used when you define standard deviation (sigma

w). Be specific.

13. Page 6, line 2. The term "z-less" tends to be a very specific term used with stable atmospheres. Please define this if you think the word is needed. Same issue on Page 8, line 8.

14. Page 6, line 9. It would help the audience to use consistent units. In this paper, most readers would really prefer that you use units such as umol/m2/s throughout. The fertilizer community often uses mass of N, but mass units really don't help this paper (and you use mass of N2O, not N). In this particular line, we are given a concentration in uL/L and then you switch to gradient of mg mˆ-4.

15. Page 6, line 24. Should not say w'; this would mean the variance of the deviation.

16. Page 8, line 26. The literature reported in Table 1 is quite selective. Please tell us why you chose these specific papers.

17. Page 9, line 11. You say "generally lower". Please quantify that it was about a factor of 2 to 5 less. Given this magnitude, what can you say about the possibilities of the technique? Also, most researchers gap-fill night periods using various techniques. Is your stable atmosphere measurement better than gapfilling these periods?

18. Page 9, line 30. I am confused why you think that advection of N2O from soybean would necessarily have a lower concentration at this time of year. The fertilizer applied to the corn field was much prior to your measurement period. This actually resulted in very low N2O fluxes through your measurement period, typically about 10% of the peak measurements that most researchers measure following fertilizer application.

19. Page 10, line 9 and 12. It looks like the accumulation method was a factor of 2 to 5 less than the chambers. These statements appear to mislead that they were close.

20. Table 2. The superscripts on the column labels look like powers; please just label the columns to avoid this. Also note that sigma w is standard deviation, not variance.

[Figure]

21. Table 3. Same issue with superscripts. The gradients are written as differential equations. In fact, you do not know this information; you have estimated this from finite difference measurements between heights. Please label appropriately.

22. Figure 3 (a). Is this the absolute value of the difference in wind direction? It is always positive.

23. Figure 3 (b) and (c). In other parts of the paper, you plot sigma w. But here you show variance; why?

24. Figure 4 (a). Variance is indicated on the right axis, but the units don't match.

25. Figure 6. If this is an accumulation starting at 1900, why don't the accumulations start at zero?

26. Figure 8. "h" is used for hour in most places, but now rainfall uses "hr".

---

## Referee Comment (RC3) · Anonymous Referee #3 · 18 Oct 2017

**General Comments**

This manuscript describes an application of the method of determining surface fluxes by quantifying the build-up of the emitted gas in a thin atmospheric surface layer during stable conditions. The experiment took place in a corn field, and the gases quantified were CO2 and N2O. A comparison with soil flux chamber results is presented. This method has been around for a while but, unlike other micrometeorological approaches, has not managed to enter the realm of operational methods because it appears too difficult to automate.

This manuscript does represent a nice evaluation of the feasibility of the method. It

is well-structured and logically consistent, and deals with the identification of stable periods in a thorough manner. However, I do have a few comments on some things that can be improved upon before publication, and some that should be considered in future similar studies.

The approach to determining 6.3m as being the "lid" to the surface accumulation seems a bit arbitrary and more a result of practical limitations than physical considerations. Looking at Fig. 4(c) and (linearly) extrapolating the segment from 5m to 8m, it appears that the 400 ppm line (i.e. most likely concentration above the surface layer) is reached in a remarkably narrow band between 10 and 12m. Maybe using the geometric mean of 11m and 5m (i.e. 7.4m) would be a better estimate of the depth of the accumulation layer? Nights other than 5 August should be checked to see whether this is repeatable. More points in the vertical would have helped to shed light on this; it is a shame (and puzzling) that the 3m level misbehaved the way it did.

It is also unfortunate that even though instruments were available that could have measured eddy covariance fluxes of $CO_2$ and $N_2O$, this was apparently not done. A third estimate of nocturnal emission fluxes could have been obtained by looking at windy nights through eddy covariance. The comparison between the accumulation method and the soil chambers needs to be quantified a bit better; presenting statistics in a table would be a good approach.

The comments of the first two reviewers are excellent, and in most cases I will try not to repeat what they have already pointed out.

**Specific Comments**

Page 1 Line 6: Annual emission budgets

P1L9: remove "the concentration of"

P1L26: eddy covariance is the accepted working term. A correlation only goes from -1 to +1 and has no units.

[Figure]

P2L3: consistency with hyphens

P2L6: there is a huge range of stable nocturnal boundary layer depths, so I would leave out the 100m, or say "on the order of 100m".

P2L8-10 information in the sentence is redundant

P2L10 and elsewhere, Pendall

P2L22/23: as mentioned by another reviewer, molecular diffusivity is on the order of $10^{-5}$ m$^2$/s. Turbulent eddy diffusivities can range from near-molecular up to 10's of m$^2$/s, so I would leave the $10^{-3}$ out.

P3L7: It is standard practice to provide at least one sentence on the location (even though with the map in Fig. 1 it only takes a minute to find the place).

P4L4: Obukhov

P4L15, P6L24: remove the ' over w.

P4L22 state the Schmidt number, if a constant was used

P5L7: were these instruments cross-calibrated with the real-time instruments?

P5L21: "at 8m" duplicated

P6L5: see general comments. Seems like a rather arbitrary approach.

P6L19: should this be 2.8m?

P9: this section would be aided greatly by a table comparing the statistics of chamber vs. mass accumulation (averages, ranges, correlation coefficients etc.)

Tables 2,3: as mentioned by another reviewer, definitely change the footnote numbers, which currently look like exponents

Fig. 2: presumably the x-axis is LT?

Fig. 3: a precise definition for the change in wind direction is required. Why is it always positive? The overlap between the horizontal variance and wind direction points is a bit messy. It might be preferable to overlap the two variances.

Fig. 4: wrong units on the vertical variance
* * *

---

## Author Comment (AC4) · 13 Dec 2017

Note that the figures are labeled wrong. They are in order of corrections requested

---

## Author Response (AR1)

Observations of turbulence and gas concentrations over a flat, agricultural terrain are analysed in this manuscript and show that gas accumulation in the nocturnal boundary layer can provide reasonable estimates of $CO_2$ and $N_2O$ emissions. The site, meteorological conditions, and measurement installations were ideal for this approach. The results clearly show potential and limitations of the technique. In this sense, the study makes a useful contribution to the journal.

The only major addition I would like to propose is a broader discussion of the technique in the context of other techniques used to estimate gas exchange between land and atmosphere. In particular, I would like to see a comparison with the eddy covariance and the radon mass balance techniques (e.g. Biraud et al., 2002, Tellus, 54B, 41-60) in terms of their precision and the scale of the observed 'footprint'. As indicated below, an eddy covariance component to the project was not possible but would be helpful in future efforts to evaluate the method. The fetch indicated study (Biraud et al, 2002) cannot be used in the current study since: 1) they assumed wind flowing in a constant direction while the flow during the night events presented here clearly changed direction commonly, 2) the mean wind used in their study were on the order of 5 m/s while our study worked in a BL with mean wind speeds of 1 m/s, 3) their study is based on the entire atmospheric boundary layer (ABL) not the surface boundary layer (SBL) and as such accumulation near the surface in teir study was assumed equivalent for all gases while we observed different accumulations for $N_2O$ and $CO_2$, 4) because of the depth of the BL,the rate concentration increase at the surface and the averaging period (and lag periods) are much longer than 1.5 hours, 5) the study is based on synoptic scale events and turbulence scales not local surface boundary layer turbulence scales, 6) as a result of the ABL framework and selection of daytime and nighttime events based on synoptic conditions, the events include both unstable and stale conditions within the entire ABL, 5) as a consequence of the time scales and dominance of daytime instability, the 'footprint' is much larger than the present study timescales, winds, and stability conditions, and 6) the depth of the ABL and the footprint dimension will result in a precision that is not relevant to nocturnal SBL emissions estimates.
In agreement: Increased time duration from 2 to 12 h was found by Biraud et al. to decrease estimated flux. We also found that increasing the period from 1.5 h to 3 h decreased the flux estimate.

Biraud, S., Ciais, P., Ramonet, M., Simmonds, V.K., Monfrey, P, O'Doherty, S, Spain, G, Jennings, S.G.:

Quantification of carbon dioxide, methane, nitrous oxide and chloroform emissions over Ireland from atmospheric

observations at Mace Head. Tellus 54B, 41-60, 2002.

Minor issues
- Title: instead ". . .using changes. . ." perhaps ". . .from changes. . ."? Ok
- Page 2, line 12: ". . .mass accumulations are reported for CO2, CH4, N2O, and H2. . ." Since H2 is consumed by soil microorganisms, I would expect H2 concentrations to decrease in the nocturnal boundary layer, not to accumulate. True, it was a depletion rather than accumulation for that gas. Omitted.
- Methods: Please show coordinates of the experimental field, or at least tell the reader in which country, near which town, it is located. Coordinates added.
- Page 3, line 30: "measured", not "measure" corrected
- Precision of reported fluxes, e.g., page 7, line 15, and Table 4: How meaningful is it to report the value of a mean flux to the second digit after the decimal point, when the standard deviation is larger than the mean itself? This does not factor in to any of the rules of significant digits I know of. However, I have reduced the siginificant digits at Page 7 and on gradients and N2O accumulations in Table 4 in careful accordance with the rules: Since the data has 1 significant digit after the decimal point (N2O measurement error of 0.5 ug/m3, height measurement error of 0.1 m) and 1 significant digits after the decimal point (CO2 measurement error of 0.009 mg/m3, height measurement error of 0.1 m) significant digits, the flux could likewise have the same number of significant digits and the least accurate measure. The rest of the text uses fewer significant digits than possible (as described above).
- Mass accumulations, first paragraph: Were the comparable fluxes cited here done in a similar climatic region, with similar land management (e.g. N fertilisation)? I cannot determine where/what you are referring to.
- Page 8, line 29: The first sentence in this line states a trivial fact and can be deleted.OK

- Page 9, Discussion of lower N2O accumulation compared to chamber fluxes: Another possible explanation is that chamber fluxes were measured during the day, when soils tend to be warmer than during the night. Other parameters being equal, N2O flux from soil increases substantially with soil temperature. Diurnal chamber flux measurements were made during this part of the season with measurements showing very little difference. I have added the results of the short study.

**Anonymous Referee #2**

General Comments

This manuscript describes measurements on carbon dioxide and nitrous oxide concentration increases in the nocturnal stable surface layer to arrive at fluxes. The technique is not new, but the manuscript provides additional data to the scientific community. Overall, the methods are sound and the structure of the paper is appropriate. The study covers a period of low N2O fluxes, which creates some additional challenges for the measurements. Below are several specific comments that the author should address.

Specific Comments:

- Page 1, line 24. The use of the concentration change within the stable surface layer is also a "micrometeorological" technique. Your method is one of the micromet tools available. Yes. Not meant to indicate otherwise. Just was stating some of the methods that typically do not work under nocturnal conditions.
- Page 1, line 26. The community usually uses "eddy covariance" instead of "eddy correlation". My mistake. Corrected. That was the 'incorrect' term first used for the method.
- Page 2, line 21. I couldn't find that SBL was defined. Added here
- Page 2, line 22. Molecular diffusion rates are closer to 10ˆ-5. Yes. Corrected
- Page 2, line 22. Qualify that you mean typical turbulent diffusion coefficients during daytime. We can argue a wide range before we get to molecular diffusivity at night. Yes, so the estimated value has been removed.
- Page 3, line 26. I think you mean that the N2O MDL is 0.3 nL/L, not ul/L? yes, corrected.
- Page 3, line 28. The manual for this instrument suggests better than 1 uL/L; is your value related to precision or accuracy? Precision as measured.
- Page 4, line 5. The van de Wiel reference is quite recent, whereas similarity theory has been developed much earlier. Please give original references. van de Wiel reference refers to the local similarity scale, not general similarity.
- Check typographical (spelling) errors: e.g., Page 4 line 19 and line 26. Corrected line 19, could not find line 26 error.
- Page 4, line 27. Why were the chamber measurements made during the day? Can you give the audience an indication about how the chamber measurements would cycle diurnally? Recall that your comparison is with the night. Measurements were made routinely during the day. It was too expensive to hire students to work the night as well.
- Page 5, line 6. You use 30-minute chamber measurements for a relatively short period on each day. It would seem more reasonable to report the measurements on a reasonable time unit; typically umol/m2/s is used. It is misleading to scale this to units of "per day" with such a small, biased sample. I agree, however this was done to provide framework for most researchers that conduct chamber measurements- they typically report for daily flux based on one 30-min measurement during the day. Based on this and comments below, I have changed all units to umol/m2/s or nmol/m2/s.
- Page 6, line 1. Be consistent; use friction velocity instead of shear velocity here. Also, in several places, variance is used when you define standard deviation (sigma w). Be specific. Corrected in text. Standard deviation is used in the description of the flow conditions in Table 2 because of prior use in other papers. Variance is used more generally since it is a TKE component.
- Page 6, line 2. The term "z-less" tends to be a very specific term used with stable atmospheres. Please define this if you think the word is needed. Same issue on Page 8, line 8. I have removed the first reference to z-less flow and rephrased but retained the second usage as the diffusion across the 6.3m 'cap' is based on a exchange coefficient calculated assuming z-controlled flow..
- Page 6, line 9. It would help the audience to use consistent units. In this paper, most readers would really prefer that you use units such as umol/m2/s throughout. The fertilizer community often uses mass of N, but

mass units really don't help this paper (and you use mass of N2O, not N). In this particular line, we are given a concentration in uL/L and then you switch to gradient of mg m^-4. I have changed all flux units to umol/m2/s, nmol/m2/s, and equivalent accumulation units.

- Page 6, line 24. Should not say w'; this would mean the variance of the deviation. Ok. Changed
- Page 8, line 26. The literature reported in Table 1 is quite selective. Please tell us why you chose these specific papers. Including every paper would be pointless since this is not a review paper. I sought out representative studies (similar crop conditions and soils) that used good techniques.
- Page 9, line 11. You say "generally lower". Please quantify that it was about a factor of 2 to 5 less. Since some measurements are in the same range, it is hard to specify. I have included a table comparison as suggested by referee #3.
- Given this magnitude, what can you say about the possibilities of the technique? Also, most researchers gap-fill night periods using various techniques. Is your stable atmosphere measurement better than gapfilling these periods? No. The study shows only that there is similarity between the chamber method and this method. The method needs improved profiles above the 8m measured here to identify the 'cap' well and hence the volume of accumulation. More work is needed to actually say it is a nighttime gapfilling method.
- Page 9, line 30. I am confused why you think that advection of N2O from soybean would necessarily have a lower concentration at this time of year. The fertilizer applied to the corn field was much prior to your measurement period. This actually resulted in very low N2O fluxes through your measurement period, typically about 10% of the peak measurements that most researchers measure following fertilizer application. Yes, the N applications discussed did not strongly influence the emissions but the study was late in the season when little N was still available. Discussion on relative soybean emissions was based on Table 1. Fluxes were similar to maize with no N applied (Fig. 8). I have expanded on this topic.
- Page 10, line 9 and 12. It looks like the accumulation method was a factor of 2 to 5 less than the chambers. These statements appear to mislead that they were close. I have added a table (Table 5) illustrating the differences and changed the text to clarify.
- Table 2. The superscripts on the column labels look like powers; please just label the columns to avoid this. Also note that sigma w is standard deviation, not variance. Corrected
- Table 3. Same issue with superscripts. The gradients are written as differential equations. In fact, you do not know this information; you have estimated this from finite difference measurements between heights. Please label appropriately. Corrected
- Figure 3 (a). Is this the absolute value of the difference in wind direction? It is always positive. Yes, it is absolute. Now indicated in caption.
- Figure 3 (b) and (c). In other parts of the paper, you plot sigma w. But here you show variance; why? As part of the TKE.
- Figure 4 (a). Variance is indicated on the right axis, but the units don't match. Fixed
- Figure 6. If this is an accumulation starting at 1900, why don't the accumulations start at zero? Axis label fixed
- Figure 8. "h" is used for hour in most places, but now rainfall uses "hr". Fixed

**Anonymous Referee #3**

**General Comments**
This manuscript describes an application of the method of determining surface fluxes by quantifying the build-up of the emitted gas in a thin atmospheric surface layer during stable conditions. The experiment took place in a corn field, and the gases quantified were CO2 and N2O. A comparison with soil flux chamber results is presented. This method has been around for a while but, unlike other micrometeorological approaches, has not manage to enter the realm of operational methods because it appears too difficult to automate.
This manuscript does represent a nice evaluation of the feasibility of the method. It is well-structured and logically consistent, and deals with the identification of stable periods in a thorough manner. However, I do have a few comments on some things that can be improved upon before publication, and some that should be considered in future similar studies.

- The approach to determining 6.3m as being the "lid" to the surface accumulation seems a bit arbitrary and more a result of practical limitations than physical considerations. Looking at Fig. 4(c) and (linearly) extrapolating the segment from 5m to 8m, it appears that the 400 ppm line (i.e. most likely concentration above the surface layer) is reached in a remarkably narrow band between 10 and 12m. Maybe using the geometric mean of 11m and 5m (i.e. 7.4m) would be a better estimate of the depth of the accumulation layer? Nights other than 5 August should be checked to see whether this is repeatable. The lid top is somewhat arbitrary but based on the log profile of the wind giving rationale for the geometric mean within the measured range.  Since nothing is known about 11 m or anything above 8m, I cannot see a justification for 7.4m or other estimate. Using 400 ppm as the threshold is also arbitrary. I believe assuming linear gradients in extrapolation is hard to justify for a stable BL.
- More points in the vertical would have helped to shed light on this; it is a shame (and puzzling) that the 3m level misbehaved the way it did. Yes, unfortunate. An line-integrated measure would be much better next time.
- It is also unfortunate that even though instruments were available that could have measured eddy covariance fluxes of CO2 and N2O, this was apparently not done. A third estimate of nocturnal emission fluxes could have been obtained by looking at windy nights through eddy covariance. Yes, but it was not possible at the time.
- The comparison between the accumulation method and the soil chambers needs to be quantified a bit better; presenting statistics in a table would be a good approach. This was not done due to the strong tendency of decreasing flux over time. However a table was added (Table 5) representing there time periods.

- **Specific Comments**
- Page 1 Line 6: Annual emission budgets.  A budget would include sinks. Unclear what an emission budget would be.  Not changed.
- P1L9: remove "the concentration of" OK corrected
- P1L26: eddy covariance is the accepted working term. A correlation only goes from -1 to +1 and has no units. My mistake. Corrected.  That was the 'incorrect' term first used for the method.
- P2L3: consistency with hyphens Fixed
- P2L6: there is a huge range of stable nocturnal boundary layer depths, so I would leave out the 100m, or say "on the order of 100m". OK corrected
- P2L8-10 information in the sentence is redundant Since nocturnal inversions can also occur with substantial warm air advection, this description is there to indicate a radiation inversion. Retained.
- P2L10 and elsewhere, Pendall OK corrected
- P2L22/23: as mentioned by another reviewer, molecular diffusivity is on the order of $10^{-5}$ m2/s. Turbulent eddy diffusivities can range from near-molecular up to 10's of m2/s, so I would leave the $10^{-3}$ out. Agreed. Removed.
- P3L7: It is standard practice to provide at least one sentence on the location (even though with the map in Fig. 1 it only takes a minute to find the place). Fixed.
- P4L4: Obukhov OK corrected
- P4L15, P6L24: remove the ' over w. OK, though should not matter
- P4L22 state the Schmidt number, if a constant was used 0.91 for $CO_2$ and 0.95 for $N_2O$- added
- P5L7: were these instruments cross-calibrated with the real-time instruments? No- Gas chromatograph samples too small to get equivalent real-time measurements without affecting pressure.
- P5L21: "at 8m" duplicated
- P6L5: see general comments. Seems like a rather arbitrary approach. Yes, it is.  See comments above. As it stands, the magnitudes of the accumulation are obviously sensitive to the height.  I have added this statement to the results and conclusions.
- P6L19: should this be 2.8m?
- P9: this section would be aided greatly by a table comparing the statistics of chamber vs. mass accumulation (averages, ranges, correlation coefficients etc.) Since chamber emissions decline throughout the period, a table with chamber measurements would be problematic.  This is why I did not do so.
- Tables 2,3: as mentioned by another reviewer, definitely change the footnote numbers, which currently look like exponents This has been revised to clarify
- Fig. 2: presumably the x-axis is LT? Yes, added to caption

- Fig. 3: a precise definition for the change in wind direction is required. Why is it always positive? The overlap between the horizontal variance and wind direction points is a bit messy. It might be preferable to overlap the two variances. Wind direction differences are absolute values.  It is now indicated in the caption and axis label. I have shifted the axis a bit to remove most overlap
- Fig. 4: wrong units on the vertical variance Corrected

[revised manuscript text omitted]
 | δΔTs/Δdz (°C m⁻¹) | ΔdN₂O/Δdz (μgmol m⁻⁴) | ΔdCO₂/Δdz (mgmol m⁻⁴) | K N2O (m² s⁻¹) | KCO2 (m² s⁻¹) |
|---|---|---|---|---|---|---|---|
| 1900-0300 | | Mean | -0.008 | 0.083.26 | 1.1144.4 | 0.008 | 0.008 |
| | Low turbulence u*⁺⁺<=0.05 ms⁻¹ | Standard deviationSD⁴ | 0.033 | 0.145.50 | 1.2349.1 | 0.024 | 0.022 |
| | | Mean | 0.148 | 0.00.054 | 0.218.2 | 0.233 | 0.221 |
| | Turbulent u*>0.05 ms⁻¹ | Standard deviationSD | 0.025 | 0.093.40 | 0.3815.3 | 0.229 | 0.216 |
| 0300-0700 | | Mean | 0.005 | 0.062.22 | 0.8232.6 | 0.010 | 0.009 |
| | Low turbulence u*<=0.05 ms⁻¹ | Standard deviationSD | 0.053 | 0.093.55 | 0.9638.3 | 0.111 | 0.105 |
| | | Mean | 0.270 | 0.00-0.12 | 0.020.6 | 0.601 | 0.568 |
| | Turbulent u*>0.05 ms⁻¹ | Standard deviationSD | 0.035 | 0.197.61 | 0.187.0 | 0.307 | 0.290 |

+1: u* =friction velocity
+2: Ts =sonic temperature
+3: K =diffusion coefficient
4: SD=standard deviation

[revised manuscript text omitted]

---

## Editor Decision (ED1)

**Editor Comments to revised manuscript amt-2017-278**

In the first part, referee comments are listed that have not been answered by the authors in a satisfactory way. Original referee comments (RC) and author responses (AR) are shown in blue. The new editor comments (EC) are printed in black.
In the second part, additional editor comments are listed that are not directly related to referee comments.

**RC(#1):** The only major addition I would like to propose is a broader discussion of the technique in the context of other techniques used to estimate gas exchange between land and atmosphere. In particular, I would like to see a comparison with the eddy covariance and the radon mass balance techniques (e.g. Biraud et al., 2002, Tellus, 54B, 41-60) in terms of their precision and the scale of the observed 'footprint'.
*AR: As indicated below, an eddy covariance component to the project was not possible but would be helpful in future efforts to evaluate the method. The fetch indicated study (Biraud et al, 2002) cannot be used in the current study since: 1) they assumed wind flowing (......)*
**EC 1)** The authors gave some detailed consideration to this referee comment in their response, but did not provide any additional discussion in the revised manuscript. This is not satisfactory. As suggested by the referee, additional discussion should be included in the manuscript on the precision and the footprint of the mass accumulation method compared to other common micrometeorological methods.

**RC(#1):** Mass accumulations, first paragraph: Were the comparable fluxes cited here done in a similar climatic region, with similar land management (e.g. N fertilisation)?
*AR: I cannot determine where/what you are referring to.*
**EC 2)** This referee comment obviously refers to the first paragraph of Section 3.2, where the results of the present study are compared to Wagner-Riddle et al. (2007), Venterea et al. (2005) and Mosier et al. (2006). Please address the reviewer comment in the manuscript text.

**RC(#1):** Page 9, Discussion of lower N2O accumulation compared to chamber fluxes: Another possible explanation is that chamber fluxes were measured during the day, when soils tend to be warmer than during the night. Other parameters being equal, N2O flux from soil increases substantially with soil temperature.
*AR: Diurnal chamber flux measurements were made during this part of the season with measurements showing very little difference. I have added the results of the short study.*
**EC 3)** Despite the short additional study with diurnal/diel chamber measurements, the first sentence of Section 2 is still misleading ("...using three methods during the night between 2000 and 0400 local time...") because the main part of the chamber measurements were made during the day. Please reformulate this sentence to make it more accurate.

**RC(#2):** 14. Page 6, line 9. It would help the audience to use consistent units. In this paper, most readers would really prefer that you use units such as umol/m2/s throughout. The fertilizer community often uses mass of N, but mass units really don't help this paper (and you use mass of N2O, not N). In this particular line, we are given a concentration in uL/L and then you switch to gradient of mg m^-4.
*AR: I have changed all flux units to umol/m2/s, nmol/m2/s, and equivalent accumulation units.*
**EC 4)** I agree with the revised units. However, at some positions in the manuscript the old units still need to be changed: e.g. in the abstract and in Section 3.1. Please check the entire manuscript for consistent units.

**RC(#2):** 16. Page 8, line 26. The literature reported in Table 1 is quite selective. Please tell us why you chose these specific papers.
*AR: Including every paper would be pointless since this is not a review paper. I sought out*

*representative studies (similar crop conditions and soils) that used good techniques.*
**EC 5)** Include a statement about the literature selection in the text or in the Table caption.

**RC(#2):** 19. Page 10, line 9 and 12. It looks like the accumulation method was a factor of 2 to 5 less than the chambers. These statements appear to mislead that they were close.
*AR: I have added a table (Table 5) illustrating the differences and changed the text to clarify.*
**EC 6)** The introduction of Table 5 with a comparison of time dependent averages is a useful improvement of the manuscript. However, I have difficulties to identify some of the averages in Table 5 with the data points shown in Figs. 7 and 8. It should be clarified, which data sets (symbols/colors in Figs. 7/8) were included in the calculation of the Table 5 averages. Moreover the use of the term "comparable" for the indication of agreement between the different flux methods is very non-specific and non-quantitative (e.g. Page 9, line 11/13; Page 10, line 15 in the revised version). This should be rephrased in a better way.

**RC(#2):** 25. Figure 6. If this is an accumulation starting at 1900, why don't the accumulations start at zero?
*AR: Axis label fixed*
**EC 7)** The renaming of the y-axis labels to "Accumulation domain ..." is confusing and does not satisfactorily address the referee comment. This needs to be improved.

**RC(#3):** Page 1 Line 6: Annual emission budgets.
**AR:** A budget would include sinks. Unclear what an emission budget would be.  Not changed.
**EC 8)** I think the referee wanted to point out that some greenhouse and other gases show a bi-directional behavior (unusual but sometimes occurring also for N2O). This should be considered with such a general statement.

**RC(#3):** P4L22 state the Schmidt number, if a constant was used
*AR: 0.91 for CO2 and 0.95 for N2O- added*
**EC 9)** Given that the observed $K_c$ values listed in Table 3 are much larger than the molecular diffusion limit, the turbulent Schmidt numbers rather than molecular Schmidt numbers should be used here!? Please explain.

**RC(#3):** P5L21: "at 8m" is duplicated
**EC 10)** Still needs to be corrected (at Page 5, line 29 in the revised version)

ADDITIONAL EDITOR COMMENTS
(page and line numbers refer to the revised manuscript version)

**EC 11)** Page 1, line 13-14: "Fluxes during calm nights …". It is not clear which flux results are presented here, because in the previous sentences both accumulation and diffusive fluxes have been mentioned. Please specify.

**EC 12)** Page 1, line 26: Why is the 'integrated horizontal mass flux method' not usable under calm conditions? It does not use any turbulence parameters.

**EC 13)** Page 2, line 21-25: It should be mentioned that a pure molecular diffusion (without intermittent turbulence) virtually never happens in the real world surface layer (as also indicated by the $K_c$ results in Table 3).

**EC 14)** Page 3, line 13-17. It is quite difficult for the reader to understand on which fields the described fertilisations have been applied. Moreover, the information in line 17 ("spring pre-plant application of 200 kg/ha on a field directly south") seems to contradict the information on Page 9, line 4 ("The field south of the tower, on which no N was applied during the year

…"). Therefore please clearly mark and number the mentioned fields in Fig. 1 (maybe the image section has to be adjusted for that) to illustrate their relevance for the present results.

**EC 15)** Page 3, line 19: Here a lower measurement height of 2.8 m is specified while later in the text (Page 4, line 9) a height of 3 m is indicated. Which one is correct?
Please make sure that coherent values for the measurement heights are used throughout the entire manuscript.

**EC 16)** Page 4, line 16: How did you determine the MDL for $u_*$? In my experience, this MDL value is too small. Like the trace gas fluxes, also $u_*$ (and $K_c$) determined by eddy covariance is affected by large uncertainties under calm (intermittent turbulent) conditions. Therefore the uncertainty of $u_*$ is mainly limited by intermittency (non-stationarity) rather than by sensor precision. This issue needs to be discussed.

**EC 17)** Page 4, line 29-30: I assume that the reference to Mosier et al. (2006) refers to the "vented static chamber method" and not to the "two months of measurements"!? If true, the reference should be placed earlier in the sentence.

**EC 18)** Entire text: the use of the concentration units $nLL^{-1}$, $\mu LL^{-1}$ is very unusual in the trace gas flux literature. It is also not a SI unit and not consistent with the flux units used in the manuscript. I would recommend to use $nmol\ mol^{-1}$ (or ppb) and $\mu mol\ mol^{-1}$ (or ppm) instead, without need to change the numerical values.

**EC 19)** Page 7, line 18 and Page 8, line 7: In the available pdf-document, some letters/symbols are not displayed properly.

**EC 20)** Page 8, line 11: Correct reference to "Mahrt"

**EC 21)** Page 8, line13: "with or without estimated diffusion" is somewhat confusing. Better use the formulation in Table 4: "with or without measurable diffusion".
In addition the statement in this sentence needs some explanation.

**EC 22)** Page 9, line 8: Wrong section numbering. Change to "3.4"

**EC 23)** Page 9, line 25: Why "above-canopy accumulation flux"? According to Eq. 1, the accumulation flux was calculated within and above the canopy!

**EC 24)** Page 10, line 14: The second part of the sentence lacks a verb.

**EC 25)** Page 10, line 16/17: How could there be unstable conditions, when at the same time the diffusive fluxes were not measurable and the temperature gradients (acc. to Table 3) were positive? Do you consider the observed stability sign as significant?

**EC 26)** Table 2 footnotes: $\Lambda$ is the 'Obukhov length' (not stability, which would be $z/\Lambda$)

**EC 27)** Table 4: Obviously the original gradients in the $3^{rd}/4^{th}$ column have been replaced by gradient fluxes in the revised version. Accordingly the header should be changed to "Gradient flux at ...". Also the indicated units in these two columns seem to be wrong (not appropriate for indicated values). Why are the CO2 gradient fluxes so large? Please use appropriate and consistent flux units in Table 4.

**EC 28)** Fig. 7&8: It is unclear, where the chamber measurements labelled "North fld", "South fld", and "no N" have been performed. Please explain and indicate the position of the different chamber measurements in Fig. 1b.

---

## Author Response (AR2)

**Author responses to Editor Comments of revised manuscript amt-2017-278**

In the first part, referee comments are listed that have not been answered by the authors in a satisfactory way. Original referee comments (RC) and author responses (AR) are shown in blue. The new editor comments (EC) are printed in black.
In the second part, additional editor comments are listed that are not directly related to referee comments. Author comments to EC are in UPPER CASE RED.

**RC(#1):** The only major addition I would like to propose is a broader discussion of the technique in the context of other techniques used to estimate gas exchange between land and atmosphere. In particular, I would like to see a comparison with the eddy covariance and the radon mass balance techniques (e.g. Biraud et al., 2002, Tellus, 54B, 41-60) in terms of their precision and the scale of the observed 'footprint'.
*AR: As indicated below, an eddy covariance component to the project was not possible but would be helpful in future efforts to evaluate the method. The fetch indicated study (Biraud et al, 2002) cannot be used in the current study since: 1) they assumed wind flowing (..)*
**EC 1)** The authors gave some detailed consideration to this referee comment in their response, but did not provide any additional discussion in the revised manuscript. This is not satisfactory. As suggested by the referee, additional discussion should be included in the manuscript on the precision and the footprint of the mass accumulation method compared to other common micrometeorological methods. I HAVE EXPANDED DISCUSSION OF FOOTPRINT ISSUES AND EFFORTS IN LITERATURE INCLUDING BIRAUD.

**RC(#1):** Mass accumulations, first paragraph: Were the comparable fluxes cited here done in a similar climatic region, with similar land management (e.g. N fertilisation)?
*AR: I cannot determine where/what you are referring to.*
**EC 2)** This referee comment obviously refers to the first paragraph of Section 3.2, where the results of the present study are compared to Wagner-Riddle et al. (2007), Venterea et al. (2005) and Mosier et al. (2006). Please address the reviewer comment in the manuscript text. SINCE BOTH VENTEREA AND MOSIER ARE BASED ON CHAMBER MEASUREMENTS, I HAVE REMOVED THEM FROM THIS DISCUSSION. I HAVE EXPANDED THE DISCUSSION OF THE WAGNER-RIDDLE RESULTS IN THE TEXT.

**RC(#1):** Page 9, Discussion of lower N2O accumulation compared to chamber fluxes: Another possible explanation is that chamber fluxes were measured during the day, when soils tend to be warmer than during the night. Other parameters being equal, N2O flux from soil increases substantially with soil temperature.
*AR: Diurnal chamber flux measurements were made during this part of the season with measurements showing very little difference. I have added the results of the short study.*
**EC 3)** Despite the short additional study with diurnal/diel chamber measurements, the first sentence of Section 2 is still misleading (".using three methods during the night between 2000 and 0400 local time.") because the main part of the chamber measurements were made during the day. Please reformulate this sentence to make it more accurate. ACTUALLY MOST MEASURMENTS WERE MADE DURING THE NIGHT (MY MISTAKE IN THE REVISION). I HAVE EXPANDED ON THE TIMING OF CHAMBER MEASUREMENTS IN THE TEXT HERE AND IN THE METHODS.

**RC(#2):** 14. Page 6, line 9. It would help the audience to use consistent units. In this paper, most readers would really prefer that you use units such as umol/m2/s throughout. The fertilizer community often uses mass of N, but mass units really don't help this paper (and you use mass of N2O, not N). In this particular line, we are given a concentration in uL/L and then you switch to gradient of mg m^-4.
*AR: I have changed all flux units to umol/m2/s, nmol/m2/s, and equivalent accumulation units.*
**EC 4)** I agree with the revised units. However, at some positions in the manuscript the old units still need to be changed: e.g. in the abstract and in Section 3.1. Please check the entire manuscript for consistent units. CORRECTED

**RC(#2):** 16. Page 8, line 26. The literature reported in Table 1 is quite selective. Please tell us why you chose these specific papers.
*AR: Including every paper would be pointless since this is not a review paper. I sought out representative studies (similar crop conditions and soils) that used good techniques.*
**EC 5)** Include a statement about the literature selection in the text or in the Table caption.
MORE DETAILS ADDED IN THE TEXT AND ADDITIONAL LITERATURE ADDED TO TABLE

**RC(#2):** 19. Page 10, line 9 and 12. It looks like the accumulation method was a factor of 2 to 5 less than the chambers. These statements appear to mislead that they were close.
*AR: I have added a table (Table 5) illustrating the differences and changed the text to clarify.*
**EC 6)** The introduction of Table 5 with a comparison of time dependent averages is a useful improvement of the manuscript. However, I have difficulties to identify some of the averages in Table 5 with the data points shown in Figs. 7 and 8. It should be clarified, which data sets (symbols/colors in Figs. 7/8) were included in the calculation of the Table 5 averages.
THE COLUMNS IN TABLE 5 WERE REVERSED- THIS IS FIXED. EXTREME OUTLIER VALUES THAT DID NOT APPEAR IN THE FIGURES WERE DRIVING SOME OF THE TABLE RESULTS WHICH CAUSED INCONSISTENCIES: THESE OUTLIER VALUES HAVE NOW BEEN EXCLUDED FROM ALL ANALYSIS AND SO STATED IN THE TEXT. CONSEQUENTLY VALUES IN TABLE 4 HAVE CHANGED WHICH HAS PROPOGATED INTO THE TEXT IN MANY PLACES. CLARIFICATIONS HAVE BEEN MADE IN THE FIGURE 7 AND 8 CAPTIONS TO INDICATE THE DATA USED.
Moreover the use of the term "comparable" for the indication of agreement between the different flux methods is very non-specific and non-quantitative (e.g. Page 9, line 11/13; Page 10, line 15 in the revised version). This should be rephrased in a better way. REPHRASED

**RC(#2):** 25. Figure 6. If this is an accumulation starting at 1900, why don't the accumulations start at zero?
*AR: Axis label fixed*
**EC 7)** The renaming of the y-axis labels to "Accumulation domain ..." is confusing and does not satisfactorily address the referee comment. This needs to be improved. CHANGED

**RC(#3):** Page 1 Line 6: Annual emission budgets.
**AR:** A budget would include sinks. Unclear what an emission budget would be. Not changed.
**EC 8)** I think the referee wanted to point out that some greenhouse and other gases show a bi-directional behavior (unusual but sometimes occurring also for N2O). This should be considered with such a general statement. NOW I SEE. CHANGED.

**RC(#3):** P4L22 state the Schmidt number, if a constant was used
*AR: 0.91 for CO2 and 0.95 for N2O- added*
**EC 9)** Given that the observed $K_c$ values listed in Table 3 are much larger than the molecular diffusion limit, the turbulent Schmidt numbers rather than molecular Schmidt numbers should be used here!? Please explain. EXPANDED ON

**RC(#3):** P5L21: "at 8m" is duplicated
**EC 10)** Still needs to be corrected (at Page 5, line 29 in the revised version) CORRECTED

ADDITIONAL EDITOR COMMENTS
(page and line numbers refer to the revised manuscript version)

**EC 11)** Page 1, line 13-14: "Fluxes during calm nights **.**". It is not clear which flux results are presented here, because in the previous sentences both accumulation and diffusive fluxes have been mentioned. Please specify. CLARIFIED

**EC 12)** Page 1, line 26: Why is the 'integrated horizontal mass flux method' not usable under calm conditions? It does not use any turbulence parameters. THIS HAS BEEN CLARIFIED

**EC 13)** Page 2, line 21-25: It should be mentioned that a pure molecular diffusion (without intermittent turbulence) virtually never happens in the real world surface layer (as also indicated by the $K_c$ results in Table 3). THIS HAS BEEN STATED

**EC 14)** Page 3, line 13-17. It is quite difficult for the reader to understand on which fields the described fertilisations have been applied. Moreover, the information in line 17 ("spring pre-plant application of 200 kg/ha on a field directly south") seems to contradict the information on Page 9, line 4 ("The field south of the tower, on which no N was applied during the year ."). Therefore please clearly mark and number the mentioned fields in Fig. 1 (maybe the image section has to be adjusted for that) to illustrate their relevance for the present results. CLARIFICATIONS IN THE TEXT AND ADDITION MADE TO FIGURE 1.

**EC 15)** Page 3, line 19: Here a lower measurement height of 2.8 m is specified while later in the text (Page 4, line 9) a height of 3 m is indicated. Which one is correct?
Please make sure that coherent values for the measurement heights are used throughout the entire manuscript. CORRECTED

**EC 16)** Page 4, line 16: How did you determine the MDL for $u_*$? In my experience, this MDL value is too small. Like the trace gas fluxes, also $u_*$ (and $K_c$) determined by eddy covariance is affected by large uncertainties under calm (intermittent turbulent) conditions. Therefore the uncertainty of $u_*$ is mainly limited by intermittency (non-stationarity) rather than by sensor precision. This issue needs to be discussed. ASSUMED 1 COMPONENT SONIC SPEED ERROR OF 0.01 MS$^{-1}$. PROPOGATED ERROR THROUGH USTAR CALCULATION YIELDS 0.014 MS$^{-1}$. DISCUSSION ADDED

**EC 17)** Page 4, line 29-30: I assume that the reference to Mosier et al. (2006) refers to the "vented static chamber method" and not to the "two months of measurements"!? If true, the reference should be placed earlier in the sentence. CORRECTED

**EC 18)** Entire text: the use of the concentration units nLL$^{-1}$, µLL$^{-1}$ is very unusual in the trace gas flux literature. It is also not a SI unit and not consistent with the flux units used in the manuscript. I would recommend to use nmol mol$^{-1}$ (or ppb) and µmol mol$^{-1}$ (or ppm) instead, without need to change the numerical values. CORRECTED

**EC 19)** Page 7, line 18 and Page 8, line 7: In the available pdf-document, some letters/symbols are not displayed properly. CORRECTED.  ALSO CHANGED UNITS IN FIGURES 2, 4, 5

**EC 20)** Page 8, line 11: Correct reference to "Mahrt" CORRECTED

**EC 21)** Page 8, line13: "with or without estimated diffusion" is somewhat confusing. Better use the formulation in Table 4: "with or without measurable diffusion".
In addition the statement in this sentence needs some explanation. CORRECTED

**EC 22)** Page 9, line 8: Wrong section numbering. Change to "3.4" CORRECTED

**EC 23)** Page 9, line 25: Why "above-canopy accumulation flux"? According to Eq. 1, the accumulation flux was calculated within and above the canopy! CORRECTED- WAS MEANT TO DIFFERENTIATE FROM ROOT/SOIL.

**EC 24)** Page 10, line 14: The second part of the sentence lacks a verb. CORRECTED

**EC 25)** Page 10, line 16/17: How could there be unstable conditions, when at the same time the diffusive fluxes were not measurable and the temperature gradients (acc. to Table 3) were positive? Do you consider the observed stability sign as significant? DELETED, REFERENCES WERE TO WITHIN THE LAYER.

**EC 26)** Table 2 footnotes: $\Lambda$ is the 'Obukhov length' (not stability, which would be $z/\Lambda$) CORRECTED

**EC 27)** Table 4: Obviously the original gradients in the 3$^{rd}$/4$^{th}$ column have been replaced by gradient fluxes in the revised version. Accordingly the header should be changed to "Gradient flux at ...". Also the indicated units in these two columns seem to be wrong (not appropriate for indicated values). Why are the CO2 gradient fluxes so large? Please use appropriate and consistent flux units in Table 4. CORRECTED. 3$^{RD}$/4$^{TH}$ COLUMNS ARE FLUXES NOT GRADIENTS. CO2 GRADEINT HAS BEEN CORRECTED. UNITS SHOULD BE RIGHT- WHEN GRADIENT (XMOL M$^{-4}$) MULTIPLIED BY EDDY EXCHANGE COFFICIENT (M$^2$S$^{-1}$) YIELDS FLUX (XMOL M$^{-2}$ S$^{-1}$). FLUX MAGNITUDES MATCH FIGURES 7, 8 AND TABLE 5.

**EC 28)** Fig. 7&8: It is unclear, where the chamber measurements labelled "North fld", "South fld", and "no N" have been performed. Please explain and indicate the position of the different chamber measurements in Fig. 1b. NOW INDICATED IN FIGURE 1

---

## Author Response (AR3)

**Response to Associate Editor Decision: Publish subject to technical corrections** (18 Feb 2018)

Comments to the Author:
In the second revised version, the authors have addressed all editor comments. However the checking of the corrections was very cumbersome due to the lack of a manuscript version with track changes. **Sorry for that- I did not see the request for a separate pdf so did not do one. This journal has processed the ms. very differently and I was confused more than once. You however have done a great job of improving the ms. Thank You.**

The manuscript is now ready for publication subject to the following (technical) corrections:

1) Page 8, line 7: old units uL/L still need to be changed. **Done**

2) Page 9, line 10: This newly introduced sentence is very similar to the one two lines below (line 12/13) and thus can be omitted. **Deleted**

3) Page 10, line 16: Correct to "of both N2O and …"; Correct to "… boundary layer in 22% and …" **Corrected**

4) Page 10, line 17-20: In this revised statement it should be clearly specified, whether the four listed effects lead to physically high diffusive fluxes or erroneously overestimated diffusive fluxes. **Rephrased to explain factors. Z-less flow rephrased and statement moved to pg 6 under Eq. 2 discussion.**

5) Page 10, line 23: Correct to "(Figs. 7, 8)" **Done**

6) Page 10, line 31: Was the indicated average calculated over all chamber fluxes (for the differently fertilised fields) including 'no N' fluxes? Please specify. **I have clarified in text.**

7) Table 5: Do the average chamber fluxes presented in Table 5 include all chamber measurements including the measurements on the non-fertilised ("no N") fields? Please specify in the Table caption. **Added in header.**

8) Fig.1: some of the labels in panel (b) are two small (e.g. "Met station" etc.) and need to be enlarged. Increase font some. **Hard to increase a lot since I cannot control the location of the label relative to the symbol. Did increase font some.**

9) Fig.7&8: The legend labels "North fld", "South fld" for the chamber measurements are not informative. It would be more relevant to relate the chamber flux data to the field labels (fertiliser regimes) displayed in Fig. 1. **Clarified.**